# Mechanism of environmentally driven conformational changes that modulate H-NS DNA-bridging activity

Ramon A van der Valk[1], Jocelyne Vreede[2], Liang Qin[1], Geri F Moolenaar[1], Andreas Hofmann[3], Nora Goosen[1], Remus T Dame[1,4]*

[1]Leiden Institute of Chemistry, Leiden University, Leiden, Netherlands; [2]Computational Chemistry, Van 't Hoff Institute for Molecular Sciences, University of Amsterdam, Amsterdam, Netherlands; [3]Institute for Theoretical Physics, University of Heidelberg, Heidelberg, Germany; [4]Centre for Microbial Cell Biology, Leiden University, Leiden, Netherlands

**Abstract** Bacteria frequently need to adapt to altered environmental conditions. Adaptation requires changes in gene expression, often mediated by global regulators of transcription. The nucleoid-associated protein H-NS is a key global regulator in *Gram*-negative bacteria and is believed to be a crucial player in bacterial chromatin organization via its DNA-bridging activity. H-NS activity in vivo is modulated by physico-chemical factors (osmolarity, pH, temperature) and interaction partners. Mechanistically, it is unclear how functional modulation of H-NS by such factors is achieved. Here, we show that a diverse spectrum of H-NS modulators alter the DNA-bridging activity of H-NS. Changes in monovalent and divalent ion concentrations drive an abrupt switch between a bridging and non-bridging DNA-binding mode. Similarly, synergistic and antagonistic co-regulators modulate the DNA-bridging efficiency. Structural studies suggest a conserved mechanism: H-NS switches between a 'closed' and an 'open', bridging competent, conformation driven by environmental cues and interaction partners.
DOI: https://doi.org/10.7554/eLife.27369.001

*For correspondence:
rtdame@chem.leidenuniv.nl

## Introduction

Although the bacterial genome is compacted by a vast variety of factors, including DNA supercoiling, and macromolecular crowding, it owes much of its organization to nucleoid-associated proteins (*Dame, 2005*; *Dame et al., 2011*; *Dillon and Dorman, 2010*; *Dorman, 2013*; *Rimsky and Travers, 2011*; *Travers and Muskhelishvili, 2005*; *Luijsterburg et al., 2008*; *Dame and Tark-Dame, 2016*). A key protein in nucleoid organization of Gram-negative bacteria is the Histone-like Nucleoid Structuring protein (H-NS). Genome-wide binding studies have revealed that H-NS binds along the genome in long patches (*Grainger et al., 2006*; *Kahramanoglou et al., 2011*; *Lucchini et al., 2006*; *Navarre, 2006*; *Oshima et al., 2006*), which have been proposed to mediate the formation of genomic loops (*Noom et al., 2007*; *van der Valk et al., 2014*). H-NS is also an important regulator of global gene expression, implied in mediating global transcriptional responses to environmental stimuli (osmolarity, pH, temperature) (*Atlung and Ingmer, 1997*), and operating as xenogeneic silencer, silencing horizontally integrated DNA (*Navarre et al., 2006*). A large fraction of *Escherichia coli* and *Salmonella* genes (5–10%) is thus regulated (usually repressed) by the action of H-NS. H-NS operation is modulated by environmental stimuli and through interplay with other proteins (*Atlung and Ingmer, 1997*; *Stoebel et al., 2008*). In solution, the H-NS protein exists as a dimer, which oligomerizes at high concentrations (*Ceschini et al., 2000*; *Spurio et al., 1997*). H-NS consists of three structural domains: a C-terminal domain responsible for DNA binding (*Shindo et al., 1995*), a N-terminal

**eLife digest** The genetic information every cell needs to work properly is encoded in molecules of DNA that are much longer than the cell itself. A key challenge in biology is to understand how DNA is organized to fit inside each cell, whilst still providing access to the information that it contains. Since the way DNA is organized can influence which genes are active, rearranging DNA plays an important role in controlling how cells behave.

In *Escherichia coli* and many other bacteria, a protein called H-NS contributes to DNA reorganization by forming or rupturing loops in the DNA in response to changes in temperature, the levels of salt and other aspects of the cell's surroundings. In controlling loop formation, it dictates whether specific genes are switched on or off. However, it remains unclear how H-NS detects the environmental changes.

To address this question, van der Valk et al. used biochemical techniques to study the activity of H-NS from *E. coli* under different environmental conditions. The experiments show that changes in the environment cause structural changes to H-NS, altering its ability to form DNA loops. A previously unnoticed region of the protein acts as a switch to control these structural changes, and ultimately affects which genes are active in the cell.

These findings shed new light on how bacteria organize their DNA and the strategies they have developed to adapt to different environments. The new protein region identified in H-NS may also be present in similar proteins found in other organisms. In the future, this knowledge may ultimately help to develop new antibiotic drugs that target H-NS proteins in bacteria.

DOI: https://doi.org/10.7554/eLife.27369.002

dimerization domain (*Bloch et al., 2003*; *Cerdan et al., 2003*; *Esposito et al., 2002*; *Ueguchi et al., 1996*) and a central dimer-dimer interaction domain responsible for multimer formation (*Arold et al., 2010*; *Leonard et al., 2009*). These two interaction domains are connected by a long α-helix (*Arold et al., 2010*) (helix α3). H-NS exhibits two seemingly distinct DNA-binding modes: DNA bridging (*Dame et al., 2006*; *Dame et al., 2000*; *Dame et al., 2001*; *Dame et al., 2002*; *Schneider et al., 2001*), the condensation of DNA by intra- and inter- molecular DNA binding by H-NS and DNA stiffening, the rigidification of DNA through the formation of a H-NS-DNA filament (*Amit et al., 2003*; *Dame and Wuite, 2003*; *Liu et al., 2010*). These modes have been attributed to the basic functional H-NS unit (a dimer) binding to DNA either in cis or *in trans* (*Wiggins et al., 2009*; *Joyeux and Vreede, 2013*). H-NS paralogues StpA, Sfh, Hfp, and truncated derivatives such as H-NST, have been proposed to modulate H-NS function by forming heteromers with H-NS (*Baños et al., 2008*; *Deighan et al., 2003*; *Müller et al., 2010*; *Williams et al., 1996*; *Williamson and Free, 2005*), with DNA-binding properties different from homomeric H-NS. Members of the Hha/YmoA family of proteins are H-NS co-regulators with limited sequence homology to H-NS (*Madrid et al., 2007*). At many targets along the genome, H-NS and Hha co-localize. Localization of Hha at these sites is strictly H-NS dependent, whereas the genome-wide binding pattern of H-NS is only mildly affected by Hha (*Ueda et al., 2013*).

Although evidence has been put forward that the concentration of divalent ions determines the binding mode of H-NS (*Liu et al., 2010*), a mechanistic explanation is lacking. Moreover, the possible effect of co-regulators of H-NS, such as Hha, on these binding modes has remained unexplored. To obtain a better understanding of the molecular basis underlying the H-NS binding modes, it is crucial to determine the effects of ion valence and concentration, as well as the presence of helper proteins on both the stiffening and the bridging mode. Here, we investigate DNA stiffening on short DNA tethers using Tethered Particle Motion (TPM). As *intramolecular* DNA bridging does not occur on short DNA tethers, DNA stiffening can be uncoupled from *DNA bridging*. In addition, to accurately determine *intermolecular* DNA-bridging efficiencies in solution, we developed a sensitive quantitative bulk assay. Using these two assays, we unravel the assembly pathway of bridged DNA-H-NS-DNA complexes and the roles of mono- and divalent ions, helper proteins Hha and YdgT, and truncated H-NS derivatives. Finally, Molecular Dynamics (MD) simulations reveal that ions and inter-acting proteins *directly* alter H-NS structure from a 'closed' bridging incapable to an 'open' bridging

capable conformation, thus providing a molecular understanding of the modulation of H-NS function.

## Results

### The role of Mg$^{2+}$and H-NS multimerization in DNA bridging and DNA stiffening

In order to dissect the role of divalent ions in the formation of bridged and stiffened complexes, we applied a novel, sensitive, and *quantitative* DNA-bridging assay and carried out TPM experiments (providing a quantitative and selective readout of DNA stiffening). The DNA-bridging assay relies on immobilization of bait DNA on magnetic microparticles and the capture and detection of $^{32}$P labeled prey DNA if DNA-DNA bridge formation occurs (see *Figure 1—figure supplement 4b* for a schematic depiction of the assay). 80% of initial prey DNA is recovered at high H-NS concentrations (see *Figure 1a*). In the absence of either the H-NS protein or bait DNA, no prey DNA is recovered under our experimental conditions. Next, we used this assay to quantify the DNA-bridging efficiency of H-NS as a function of the amount of Mg$^{2+}$ ions (see *Figure 1b*), reproducing the qualitative results of *Liu et al. (2010)* and providing independent confirmation of the previously observed effects. The concentration range from 0 to 10 mM Mg$^{2+}$ is considered to be physiologically relevant (*Hurwitz and Rosano, 1967*). Importantly, the transition from no bridging to complete bridging is abrupt between 4–6 mM Mg$^{2+}$, indicating that changes in Mg$^{2+}$ concentration might drive a binary switch.

Earlier studies have shown that H-NS binding along a single DNA molecule results in DNA stiffening (*Amit et al., 2003*; *Liu et al., 2010*). However, due to the co-occurrence of DNA bridging, previous studies were incapable of measuring DNA stiffening in the presence of Mg$^{2+}$. Here, we used TPM to investigate DNA stiffening as a function of Mg$^{2+}$ concentration. In TPM, the Root Mean Square displacement (RMS) of bead movement is a direct reflection of tether stiffness and length (*Figure 1—figure supplement 4a*). DNA-binding proteins can affect both, but previous studies have shown that the DNA contour length is not affected by binding of H-NS (*Dame et al., 2006*; *Dame et al., 2001*). Thus, an increase in stiffness due to H-NS binding translates into a higher RMS value of a DNA tether (*Figure 1d*). Here, we measured the effects of H-NS on DNA stiffness in the absence and presence of 10 mM Mg$^{2+}$ and confirmed that H-NS stiffens DNA (*Amit et al., 2003*; *Liu et al., 2010*); importantly, our experiments reveal that Mg$^{2+}$ does not affect the stiffness of the fully formed H-NS-DNA complexes at saturation, as in both conditions the RMS is the same (*Figure 1d*). Analysis of the binding characteristics using the McGhee-von Hippel equation revealed that the association constant of H-NS is somewhat reduced in the presence of Mg$^{2+}$, while cooperativity increases under these conditions (*Figure 3—figure supplement 1a,d*). The reduction in DNA-binding affinity of H-NS may be attributed to shielding of the negatively charged phosphate backbone by Mg$^{2+}$.

The cooperative binding of H-NS and DNA stiffening observed by TPM suggest that H-NS multimerizes along DNA, likely via the recently defined dimer-dimer interaction domain (*Arold et al., 2010*). Multimerization along DNA has been previously suggested (*Esposito et al., 2002*; *Williams et al., 1996*) but has never been conclusively demonstrated. To test this hypothesis, we generated a mutant, H-NS$_{Y61DM64D}$, predicted to have disrupted dimer-dimer interaction based on the H-NS$_{1-83}$ crystal structure (*Arold et al., 2010*). Size exclusion chromatography showed that this H-NS mutant indeed exists solely as a dimer in solution independent of protein concentration (*Figure 1—figure supplement 1b*), whereas wild-type H-NS forms large multimeric structures (*Figure 1—figure supplements 1a* and *Arold et al., 2010*; *Leonard et al., 2009*). The multimerization behavior of both proteins was unaffected by the presence of Mg$^{2+}$. Electrophoretic Mobility Shift Assay confirmed that the DNA binding of the H-NS mutant is intact (*Figure 1—figure supplement 2*). TPM experiments reveal that H-NS$_{Y61DM64D}$ binding does not lead to the formation of stiff H-NS-DNA filaments (*Figure 1e*). The RMS is reduced compared to that of bare DNA, indicating not only that dimer-dimer interactions are disrupted, but also that individual H-NS dimers mildly distort DNA. DNA-bridging experiments reveal that H-NS$_{Y61DM64D}$ is also incapable of forming DNA-H-NS-DNA complexes (*Figure 1—figure supplement 3*). This indicates that individual H-NS$_{Y61DM64D}$ dimers do not form stable bridges and that dimer-mediated bridging, involving H-NS-dimer binding

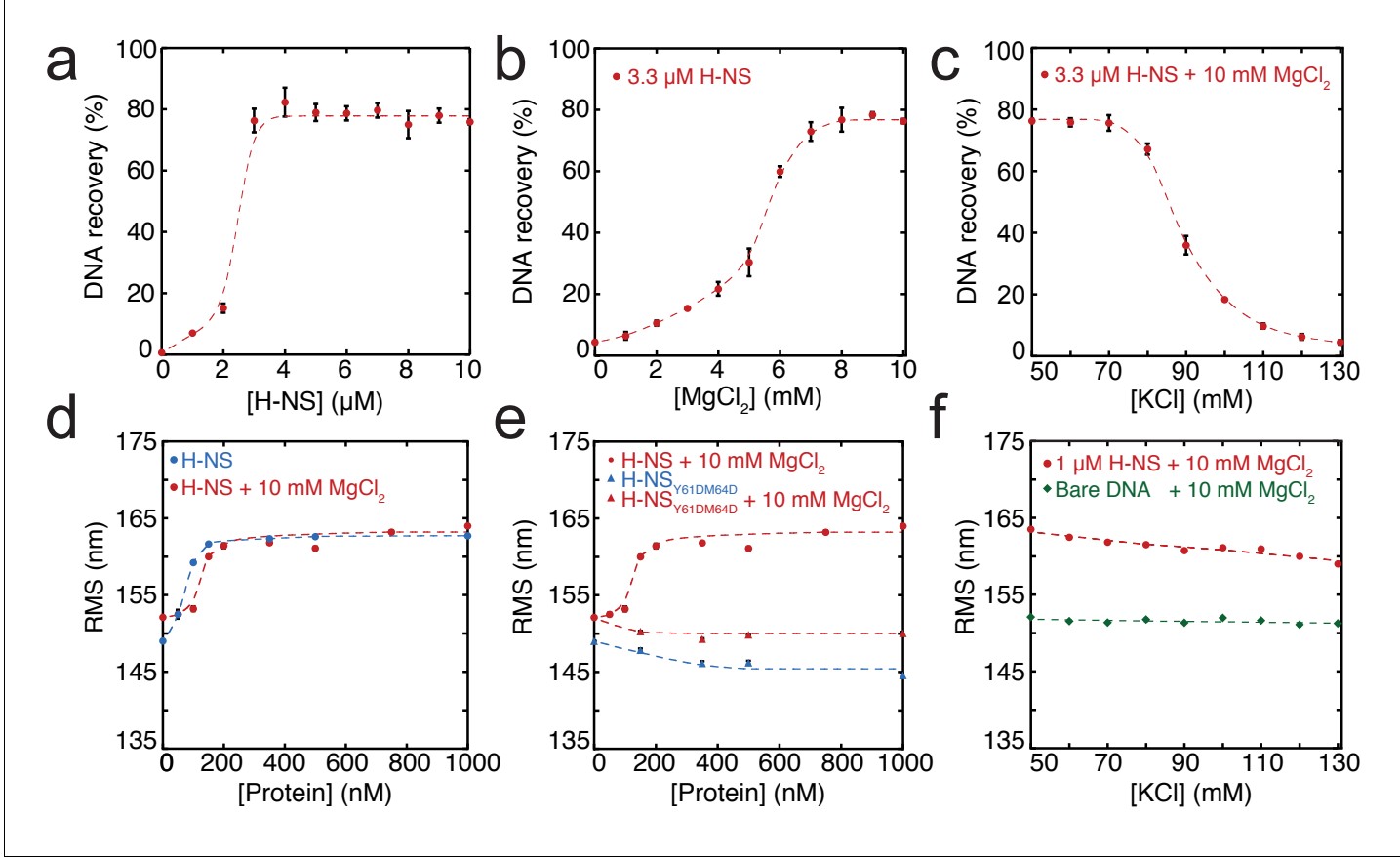

**Figure 1.** Modulation of H-NS function by ionic conditions. (a) DNA-bridging efficiency as a function of H-NS concentration in the presence of 10 mM MgCl$_2$. (b) DNA-bridging efficiency as a function of MgCl$_2$ concentration. (c) DNA-bridging efficiency as a function of the KCl concentration. (d) Root Mean Square displacement (RMS) as a function of H-NS concentration, in the presence and absence of 10 mM MgCl$_2$ (N > 70, per data point). (e) RMS of DNA as a function of H-NS$_{Y61DM64D}$ in the presence and absence of MgCl$_2$ (N > 70, for each point). (f) Extension of DNA as a function of the KCl concentration (N > 70, for each point). Error bars indicate standard deviation. Dashed lines are to guide the eye.

DOI: https://doi.org/10.7554/eLife.27369.003

The following figure supplements are available for figure 1:

**Figure supplement 1.** Multimeric state of H-NS measured using size exclusion chromatography.
DOI: https://doi.org/10.7554/eLife.27369.004

**Figure supplement 2.** Electrophoretic Mobility shift assay.
DOI: https://doi.org/10.7554/eLife.27369.005

**Figure supplement 3.** DNA recovery of H-NS and.
DOI: https://doi.org/10.7554/eLife.27369.006

**Figure supplement 4.** Schematic depiction of techniques used in this study.
DOI: https://doi.org/10.7554/eLife.27369.007

**Figure supplement 5.** Modulation of H-NS by alternative anions.
DOI: https://doi.org/10.7554/eLife.27369.008

cooperativity due to high local DNA concentration adjacent to existing bridges (*Dame et al., 2006*; *Dame et al., 2000*), is insufficient to explain the formation of bridged DNA-H-NS-DNA complexes. The nature of the effect of Mg$^{2+}$ on DNA-bridging efficiency is not understood. An increase in affinity would be expected if Mg$^{2+}$ would only facilitate interactions between the DNA phosphate backbone and negatively charged residues on H-NS, but this is not observed (*Figure 3—figure supplement 1*). As Mg$^{2+}$ does not affect the multimeric state of H-NS in solution (*Figure 1—figure supplement 1*), a model involving an effect on H-NS multimerization can be excluded. A structural effect of Mg$^{2+}$ on individual units within H-NS filaments could explain the observed effects of Mg$^{2+}$ on the bridging efficiency of H-NS.

## Mg$^{2+}$ alters H-NS structure

To investigate the role of Mg$^{2+}$ on individual H-NS dimers we carried out MD simulations of an H-NS dimer in both the absence and presence of Mg$^{2+}$, using our previously established model of a full-length H-NS dimer (van der Valk et al., 2014). Visual inspection of the H-NS dimer simulations at 50 mM KCl reveals that H-NS changes from an 'open' extended conformation into more compact 'closed' shapes (see *Figure 2a* for snapshots from these simulations, *Figure 2—figure supplement 1* for examples of the 'closed' conformation, or *Figure 2—video 1* for a movie of one such simulation). The three domains in H-NS interact, and these inter-domain interactions are facilitated by partial unfolding and buckling of the long central α helix (helix α3) connecting the dimerization and dimer-dimer interaction domains. The average distance between the donor and acceptor of all helical hydrogen bonds (O-H distance) in helix α3 indicates that the buckle forms in region Glu42-Ala49 (*Figure 2b*). By analyzing the O-H distance between residues Ser45 and Ala49 in time, key residues at the site of buckle formation (see *Figure 2—figure supplement 3*), we found that buckles can be reversible and irreversible, within the simulation time scale of 50 ns. Reversible buckles, caused by thermal fluctuations, typically last a few nanoseconds and occur several times during a single simulation run (see green line in *Figure 2—figure supplement 3* for an example). Irreversible buckles, stabilized by inter-domain interactions, do not return to a helical conformation during our simulations (see red line in *Figure 2—figure supplement 3* for an example). To characterize the interactions that occur during the simulations, we generated contact maps that show the probability of finding interactions between residues, with a contact defined as the minimum distance between two residues being 0.6 nm or less, see Materials and methods for details of this analysis (*Figure 2—figure supplement 2a*). These maps reveal that the DNA-binding domain interacts with other parts of the protein complex. In absence of Mg$^{2+}$, the DNA-binding domain interacts with the dimerization domain, rendering the DNA-binding QGR motif (residues 112–114) (Gordon et al., 2011) of one DNA-binding domain inaccessible (see the snapshots in *Figure 2a*). Although the simulations were performed in absence of DNA, these observations suggest that DNA bridging is not possible in such a conformation, as the H-NS dimer can bind DNA only through its remaining/second DNA-binding domain. In this 'closed' conformation, interactions occur between the IRT residues at position 10–12 and the AMDEQGK residues at position 122–128. These interactions are hydrophilic in nature, supplemented by a salt bridge between R11 and D124 or E125. In the presence of Mg$^{2+}$ interactions between the DNA-binding domain and the dimerization domain no longer occur (see the snapshots in *Figure 2a* or the movie in *Figure 2—video 2*), The absence of such interactions is further illustrated by the contact map in *Figure 2—figure supplement 2b* and in higher detail in *Figure 2—figure supplement 5*. The likelihood of finding Mg$^{2+}$ interacting with (i.e. being within 0.6 nm of) H-NS residues, indicated by $P_{Mg2+}$, revealed that Mg$^{2+}$ has a preference for glutamate residues in region 22–35 (*Figure 2c*, *Figure 2—figure supplement 8*), where the ions shield this region from interacting with the DNA-binding domains. The Mg$^{2+}$ ions transiently interact with the glutamate residues, with residence times in the order of a few ns (as seen in *Figure 2—video 2*). Furthermore, Mg$^{2+}$ ions interact with region 98–105 (sequence DENGE), right next to the DNA-binding QGR motif (*Figure 2c*). The presence of Mg$^{2+}$ stabilizes the 'open' conformation of H-NS, ensuring that DNA bridging can occur. Furthermore, we noted that Mg$^{2+}$ is also located close to the buckle and may directly stabilize helix α3 through interactions with Glu42, Glu43, Glu44, and Ser45 (*Figure 2—figure supplement 7*), resulting in an 'open', bridging capable, H-NS conformation (see *Figure 2—videos 1* and *2*). These data suggest that Mg$^{2+}$ modulates H-NS by shielding interactions between the DNA-binding domain and dimerization domain, and by influencing the conformation of helix α3. Based on these observations, we designed an H-NS mutant predicted to bridge DNA independent of Mg$^{2+}$. We therefore generated a mutant in which several of the amino acids involved in buckle formation (E43,E44,S45) were substituted with alanines. In our DNA-bridging assay, H-NS$_{E43A,E44A,S45A}$ retains its ability to bridge DNA, but indeed achieves high DNA recovery (±50%) in the absence of Mg$^{2+}$ (*Figure 2—figure supplement 9a*). Low concentrations of Mg$^{2+}$ are sufficient to reach saturated DNA bridging (up to 80% DNA recovery) (*Figure 2—figure supplement 9a*). This observation independently confirms our model that the stretch of glutamates at the buckle is responsible for Mg$^{2+}$ sensing and Mg$^{2+}$-dependent bridging by H-NS. Additionally, we note that the DNA-binding affinity of H-N$_{E43A,E44A,S45A}$ is not significantly altered and that its DNA binding cooperativity is similar to that of wildtype H-NS in the absence of Mg$^{2+}$ (*Figure 2—figure supplement 9b*, *Figure 3—*

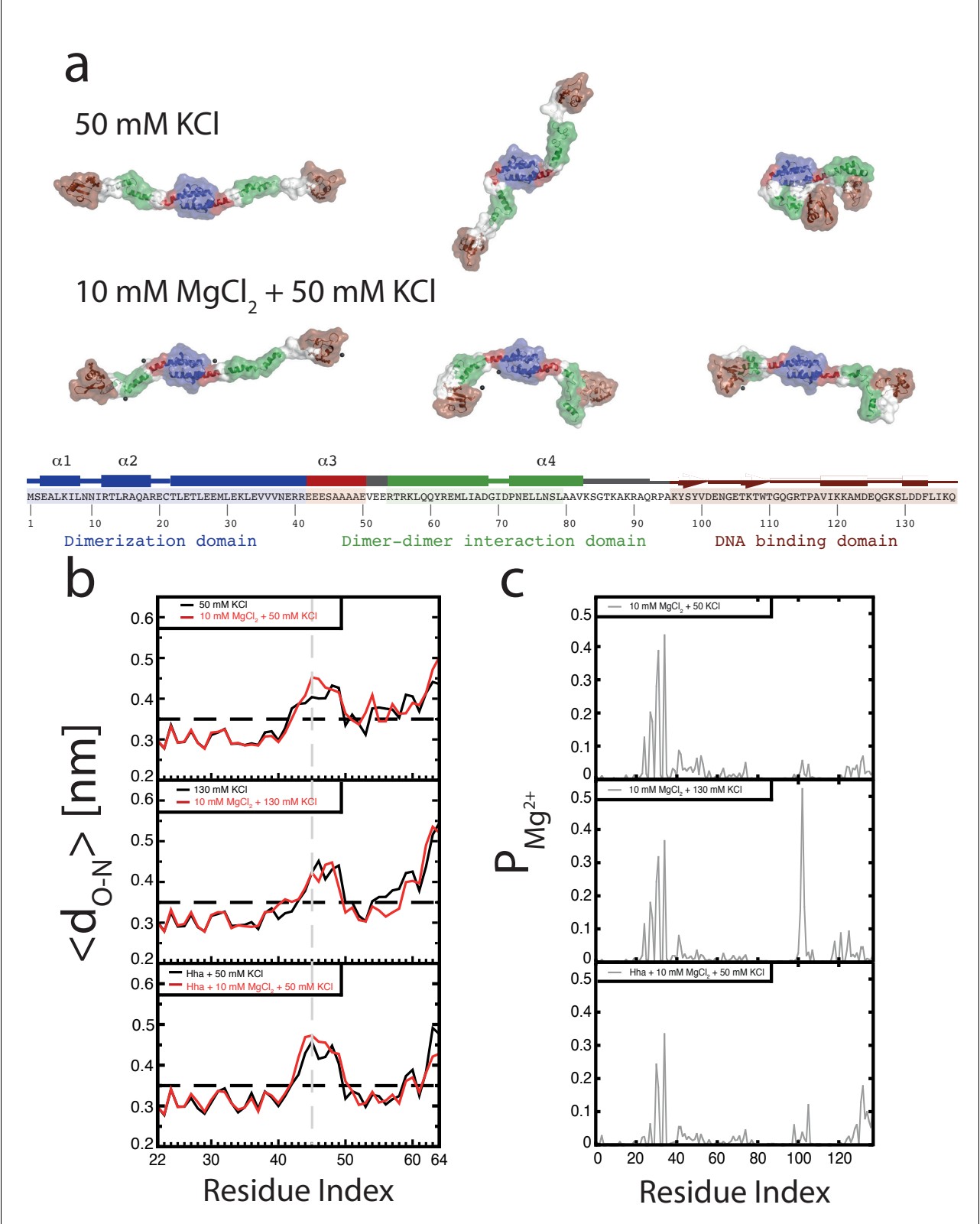

**Figure 2.** Conformation of the H-NS dimer as a function of osmolarity. (a) Snapshots depicting representative conformations of H-NS in the simulations with 50 mM KCl (top) and 10 mM MgCl₂ +50 mM KCl (bottom) (For the full movies depicting these effects see *Figure 2—videos 1* and *2*, for other examples of the 'closed' conformations of H-NS see *Figure 2—figure supplement 1*). The buckle region is highlighted in red and the domains are colored blue, green and brown for the dimerization domain, the dimer-dimer interface and the DNA-binding domain, respectively. The conserved motif
*Figure 2 continued on next page*

*Figure 2 continued*

involved in DNA binding is highlighted as ball-stick models. $Mg^{2+}$ ions are shown as dark gray orbs. The protein is shown in ribbon representation with a transparent surface. The atomic radii in the protein were set to 3 A to smooth the surface. (b) Location of buckle in helix α3. The average distance $d_{O-N}$ between donor and acceptor in the helical hydrogen bond in helix α3 is plotted as a function of the residue index of the acceptor. The dashed black line in the graphs indicates the distance threshold for forming a hydrogen bond. Time traces of these distances are given in *Figure 2—figure supplement 3*. (c) Location of $Mg^{2+}$ on H-NS. The probability of finding $Mg^{2+}$ ions within 0.6 nm of an H-NS residue, $P_{Mg^{2+}}$, is plotted as function of the residue index for the three systems containing $Mg^{2+}$.

DOI: https://doi.org/10.7554/eLife.27369.009

The following video and figure supplements are available for figure 2:

**Figure supplement 1.** Examples of 'closed' H-NS conformations in the presence of (a) 50 mM KCl, (b) 130 mM KCl, or (c) Hha.
DOI: https://doi.org/10.7554/eLife.27369.010

**Figure supplement 2.** Contact maps of full-length H-NS dimers simulations in different conditions.
DOI: https://doi.org/10.7554/eLife.27369.011

**Figure supplement 3.** Time traces of the O-H distance between residues 45 and 49.
DOI: https://doi.org/10.7554/eLife.27369.012

**Figure supplement 4.** Location of $K^+$ on hr-NS.
DOI: https://doi.org/10.7554/eLife.27369.013

**Figure supplement 5.** Contact maps of H-NS dimers in different conditions, focused on the interactions between the dimerization domain and the DNA-binding domain.
DOI: https://doi.org/10.7554/eLife.27369.014

**Figure supplement 6.** Location of Hha on H-NS.
DOI: https://doi.org/10.7554/eLife.27369.015

**Figure supplement 7.** Correlation between hydrogen bond distance and proximity of $Mg^{2+}$ to the buckle in helix α3.
DOI: https://doi.org/10.7554/eLife.27369.016

**Figure supplement 8.** $Mg^{2+}$ localization on H-NS.
DOI: https://doi.org/10.7554/eLife.27369.017

**Figure supplement 9.** Function of the H-NS derivative, $H-NS_{E43A,E44A,S45A}$.
DOI: https://doi.org/10.7554/eLife.27369.018

**Figure 2—video 1.** Conformational flexibility of H-NS.
DOI: https://doi.org/10.7554/eLife.27369.019

**Figure 2—video 2.** The effect of magnesium on the conformational flexibility of H-NS.
DOI: https://doi.org/10.7554/eLife.27369.020

---

*figure supplement 1d*). Yet, in the presence of $Mg^{2+}$, wild-type H-NS exhibits a far more cooperative DNA binding (*Figure 3—figure supplement 1d*). This suggests that the transition between the 'open' and 'closed' conformation of H-NS promotes lateral filament formation by H-NS.

## Modulation of DNA bridging by osmotic factors

Although it has long been known that the expression of some H-NS controlled genes (such as the *proU* operon) is modulated by the osmolarity of the medium (*Cairney et al., 1985*), the underlying mechanism remains undetermined. Previous studies have revealed that the H-NS DNA stiffening mode is mildly sensitive to the KCl concentration (*Amit et al., 2003*; *Liu et al., 2010*). Using TPM, we were able to confirm these observations. The reduction in DNA stiffening is gradual (*Dame and Wuite, 2003*) and modest (*Figure 1f*). It is thus questionable whether the multimerization of H-NS along DNA alone is sufficient to explain its role in repression of transcription (and modulation thereof). Could the modulation of gene repression be due to ionic effects on DNA-bridging efficiency? Using our DNA-bridging assay, we observed complete abolishment of H-NS DNA bridging by KCl at concentrations exceeding 120 mM (*Figure 1c*), a binary response, similar to what we observed for the $Mg^{2+}$ titration. This in vitro observation mirrors the in vivo response of the ProU operon, at which KCl concentrations exceeding 100 mM are required to alleviate H-NS-mediated repression (*Cairney et al., 1985*). Control experiments using K-glutamate (*Figure 1—figure supplement 5*) confirm that $K^+$, and not the counter-ion, is responsible for the observed effects, even though the counter ion may affect the DNA-binding affinity of the protein (*Figure 1—figure supplements 5c* and *Leirmo et al., 1987*) This strong and abrupt effect on DNA bridging, while leaving DNA stiffening essentially unaffected, might indicate that H-NS reverts to the 'closed' conformation by the addition of $K^+$. To investigate this effect at a structural level we performed MD simulations at

high KCl concentrations. The presence of 130 mM KCl alters interactions between the various domains in H-NS (see contact maps in *Figure 2—figure supplement 2c and d*). In addition to interactions between the DNA-binding domain and the dimerization domain, the two DNA-binding domains in the dimer interact with each other. Moreover, the DNA-binding domains interact with helix α3. In particular regions 98–105 (KYSYVDENGE) and 123–129 (EQGKS) are involved in these interactions. The average distance between the donors and acceptors in the hydrogen bonds within helix α3 (*Figure 2b*) indicates that that buckles in helix α3 also occur at a high KCl concentration, at the same location as determined by low-salt conditions. These observations indicate that the 'closed' state can have multiple forms (see *Figure 2—figure supplement 1* for snapshots), but that all these conformations block the DNA-binding motif QGR from interacting with DNA. The presence of $Mg^{2+}$ does not significantly alter the occurrence of buckles (*Figure 2b*). Instead, $Mg^{2+}$ is capable of deterring interactions between the DNA-binding domain and the dimerization domain by shielding residues in the dimerization domain involved in these interactions (*Figure 2c*, *Figure 2—figure supplement 8*). However, the probability of interactions occurring between the DNA-binding domain and other parts of the protein is reduced significantly in the presence of $Mg^{2+}$ (*Figure 2—figure supplement 2* and *Figure 2—figure supplement 5*). High K+ concentrations therefore inhibit H-NS bridging by promoting the 'closed' conformation of H-NS even in the presence of $Mg^{2+}$ and elucidates the modulatory and regulatory effects of KCl and osmolarity.

## Modulation of DNA bridging and DNA stiffening by truncated H-NS variants

In addition to environmental factors such as osmolarity, H-NS activity is affected by interactions with other proteins in vivo. Members of the Hha/YmoA protein family, such as Hha and YdgT, are known to cooperate with H-NS in repression of genes (*Baños et al., 2008*), while other proteins such as H-NST are capable of inhibiting H-NS function (*Liu et al., 2010*), likely by hampering H-NS multimerization. To systematically investigate the latter mechanism, we designed and synthesized truncated H-NS derivatives, targeting the H-NS dimerization domain (H-NS$_{1-58}$) or dimer-dimer interaction domain (H-NS$_{56-83}$). Interfering with H-NS dimerization, through the addition of H-NS$_{1-58}$ in DNA-bridging experiments, we observed a reduction in DNA recovery from 75% to 20% at ratios higher than 1: 3 H-NS$_{1-58}$/ H-NS (*Figure 3a*). Similarly, targeting the dimer-dimer interaction domain (through H-NS$_{56-82}$) resulted in complete abolishment of DNA bridging (*Figure 3a*). Next, we investigated the effects of H-NS derivatives on H-NS DNA stiffening using TPM. Only at very high H-NS derivative concentrations (30-fold excess) reduction of DNA stiffening was observed (*Figure 3b*). These experiments reveal that both H-NS dimerization and dimer-dimer interactions can be effectively targeted for inhibition of H-NS activity and that the respective domains are crucial to the formation of bridged filaments. This suggests that natural H-NS inhibitors such as H-NST operate by disrupting DNA bridging and provides clues for rational design of artificial peptide inhibitors of H-NS.

## Modulation of H-NS by Hha and YdgT

Gene regulation by H-NS often occurs in conjunction with other proteins; these co-regulators are known to interact with H-NS at specific loci along the genome. Two such proteins, Hha, and YdgT, are members of the Hha/YmoA (*Madrid et al., 2007*) family of proteins. In order to understand the modulation of H-NS function by these proteins we investigated their influence on the H-NS DNA binding modes. We observed that Hha, when added at equimolar concentrations, enhances DNA bridging by H-NS at low $Mg^{2+}$ concentrations (*Figure 3c*). A similar enhancement of H-NS-mediated DNA bridging was observed with the Hha paralogue, YdgT. While Hha and YdgT promote DNA bridging to a similar extent, YdgT promotes DNA bridging at significantly lower concentrations, likely due to a higher affinity for H-NS. At higher concentrations of YdgT the effect on H-NS-mediated bridging closely resembles the bridging profile obtained for H-NS$_{E43A,E44A,S45A}$ (*Figure 3—figure supplement 2*). To determine whether enhanced DNA bridging is due to structural changes in H-NS-DNA filaments, we investigated the effects of Hha and YdgT on DNA stiffening (*Figure 3d*). TPM experiments show a mild increase in H-NS mediated DNA stiffening in the presence of Hha. We observe a negative offset in RMS in the presence of YdgT, indicating a more compact conformation. Specifically, this is evident in TPM experiments containing H-NS, YdgT, and $Mg^{2+}$, where in the

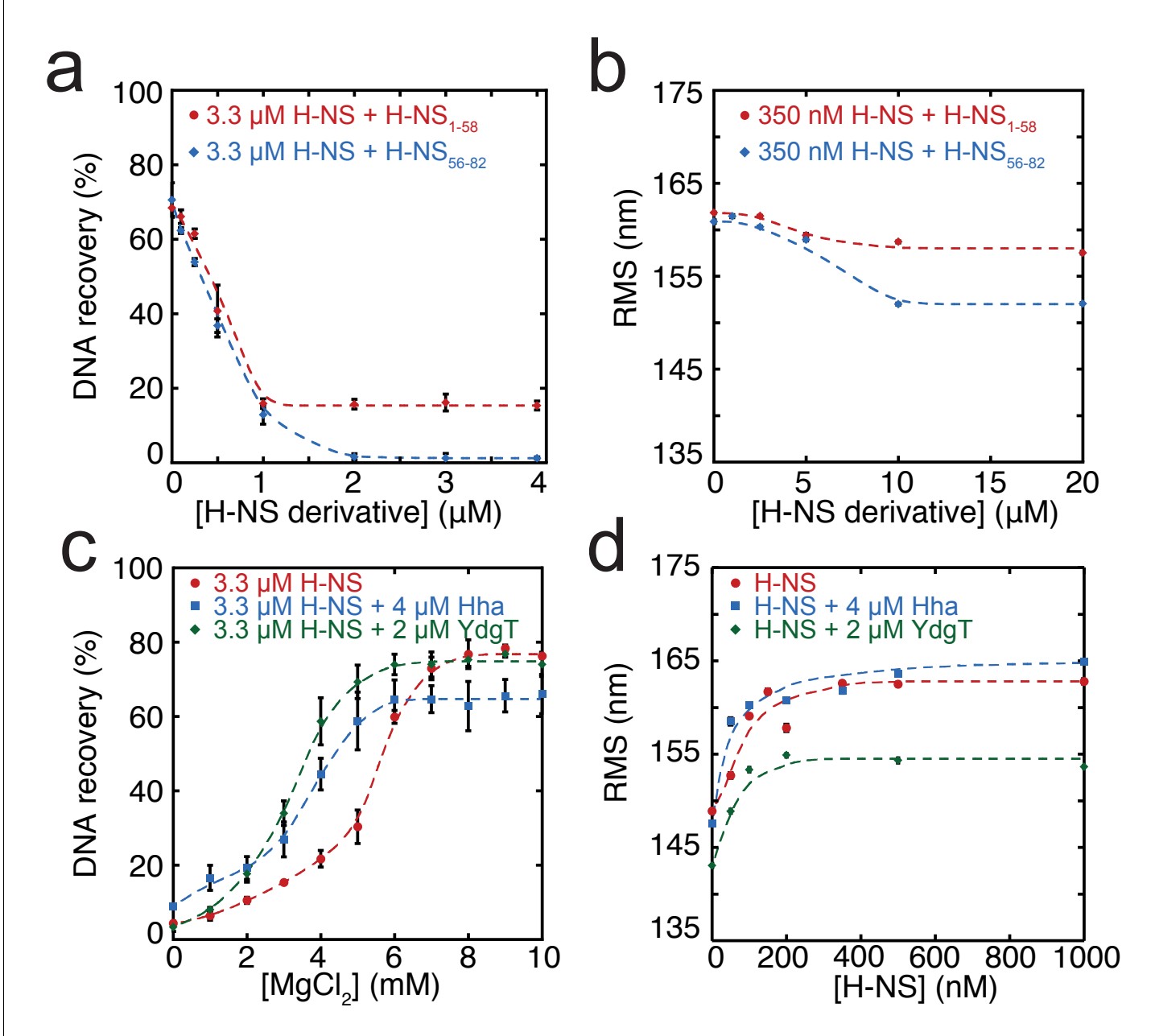

**Figure 3.** Modulation of H-NS function by protein cofactors. (a) DNA bridging efficiency as a function of inhibiting peptides targeting either the dimerization domain (H-NS$_{1-58}$) and multimerization domain (H-NS$_{56-82}$). (b) Root Mean Square displacement (RMS) as a function of inhibiting peptides targeting either the dimerization (H-NS$_{1-58}$) and multimerization (H-NS$_{56-82}$) (N > 70, for each point). (c) DNA bridging efficiency as a function of Mg$^{2+}$ concentration in the presence and absence of 4 μM Hha or 2 μM YdgT. (d) RMS of DNA in the presence of H-NS, H-NS-Hha, and H-NS-YdgT. Dashed lines are to guide the eye (N > 60, for each point).

DOI: https://doi.org/10.7554/eLife.27369.021

The following figure supplements are available for figure 3:

**Figure supplement 1.** McGhee-von Hippel analysis of H-NS DNA binding curves based on TPM data.

DOI: https://doi.org/10.7554/eLife.27369.022

**Figure supplement 2.** DNA-bridging efficiency of H-NS (red), H-NS + 4 μM YdgT (blue) and H-NS$_{E43A,E44A,S45A}$ (orange) as a function of MgCl$_2$ concentration.

DOI: https://doi.org/10.7554/eLife.27369.023

presence of Mg$^{2+}$ and YdgT, H-NS causes 'DNA collapse' in TPM (data not shown), similar to earlier observations for H-NS in the presence of Mg$^{2+}$ without (*Liu et al., 2010*; *Wang et al., 2014*) or with Hha added (*Wang et al., 2014*). This 'DNA collapse' is attributed to H-NS-mediated DNA bridge formation. One possible explanation for the effects of Hha and YdgT, is that they effectively increase the DNA-binding affinity of H-NS (*Ali et al., 2013*). To investigate whether Hha affects H-NS conformation, we performed MD simulations, incorporating structural information from the recently resolved H-NS$_{1-43}$-Hha co-crystal structure (*Ali et al., 2013*). Our MD simulations reveal that Hha does not prevent buckles in helix α3 (see *Figure 2b*). Instead Hha alters the interactions between the dimerization domain and the DNA-binding domain (*Figure 2—figure supplement 2*) by blocking access to parts of dimerization domain. This hypothesis is further supported by interactions between Hha and other parts of H-NS, including the DNA-binding domain and helix α3 (*Figure 2—figure supplement 6*). In the presence of Mg$^{2+}$ and Hha, the contacts between the DNA-binding domain and dimerization domain are reduced even further. This shows that Hha modulates H-NS function by stabilizing the 'open' -bridging capable- conformation of H-NS.

## Discussion

It has been known for many years that H-NS binding induces gene silencing. H-NS activity in vivo is modulated by physico-chemical factors (osmolarity, pH, temperature) and interaction partners. These findings support the hypothesis that H-NS plays a role in environmental adaptation. However, mechanistically it is unclear how functional modulation of H-NS by such factors is achieved. Based on our findings, we conclude that H-NS is incapable of bridging or stiffening DNA as dimers. H-NS dimers bind DNA in cis (*Dame et al., 2006*; *Dame et al., 2000*) and associate side-by-side along DNA, likely via the recently identified dimer-dimer interaction domain (*Arold et al., 2010*), resulting in DNA stiffening. This process is cooperative as H-NS dimers interact with neighbors, as well as with DNA. Our studies reveal that H-NS-DNA filaments are *structurally* very similar, independent of the presence of Mg$^{2+}$ (*Figure 1d*). But *functionally*, these H-NS-DNA filaments are distinct. H-NS can be 'activated' by Mg$^{2+}$, which promotes a conformational change, rendering both DNA-binding domains of H-NS dimers accessible for DNA bridging. In our model, the assembly of bridged complexes proceeds in distinct steps: (1) nucleation (*Lang et al., 2007*) (binding of an H-NS dimer at a high affinity site), (2) lateral filament growth by H-NS dimer-dimer interactions (leading to DNA stiffening) and (3) bridging of the assembled filament to bare DNA provided in trans (*Figure 4*). Each step can potentially be modulated by osmolarity and protein interaction partners. Here, we show that these factors most effectively target DNA bridging.

What are the implications of our observations? Our observations add to the large body of evidence showing that regulation of transcription via H-NS is complex, and that it does not proceed via a single, simple, well-defined mechanism. The most straightforward form of repression by H-NS is via occlusion of RNA polymerase from the promoter (*Lim et al., 2012*; *Prosseda et al., 2004*; *Göransson et al., 1990*). Whether this mechanism of repression involves lateral filament formation or bridging is unclear. It is expected that both types of complexes assembled at a promoter site can in principle occlude RNA polymerase. A second mechanism of repression is to trap RNA polymerase

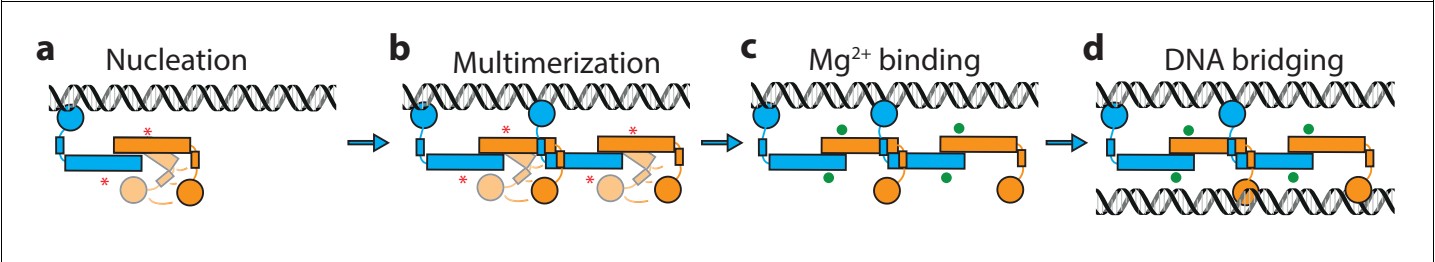

**Figure 4.** Model of H-NS complex assembly. (**a**) H-NS nucleates at preferred DNA sequences in the genome. (**b**) H-NS laterally multimerizes laterally along the DNA in the 'closed' conformation. (**c**) In the presence of Mg$^{2+}$ or other H-NS modulators such as Hha, H-NS switches to the 'open', bridging capable conformation. (**d**) H-NS forms DNA bridges in trans. The red asterisk indicates the buckle location. Mg$^{2+}$ ions are shown as green orbs.
DOI: https://doi.org/10.7554/eLife.27369.024

at the promoter, preventing promoter escape (*Schröder and Wagner, 2000*; *Shin et al., 2005*). RNA polymerase trapping likely involves DNA bridging of promoter upstream and downstream elements (*Dame et al., 2002*; *Shin et al., 2005*). A third mechanism of repression by H-NS is to interfere with RNA polymerase progression during active transcription by intragenic binding (*Dole et al., 2004*). In this model, both modes could interfere with transcription. However, it was suggested recently that only bridged filaments are capable of interfering with transcription (*Kotlajich et al., 2015*), with lateral filaments likely being disassembled as RNA polymerase encounters them. Generally, the type of complex formed by H-NS is expected to depend on the type and number of ions, combined with local DNA conformation (dependent on DNA sequence or DNA topology), and the presence of modulating proteins. The interplay between these factors will determine the strength of the complex, and degree of repression. Thus, different H-NS repressed genes are expected to be subject to different types of modulation, providing a key to a coordinated response in gene expression to altered conditions and selectivity for the interplay with specific co-regulators at specific target regions.

# Materials and methods

## Construction of expression vectors

### H-NS expression vector
The vector pRD18 for expression of H-NS was constructed by inserting the PCR amplified *hns* gene into the pET3His overexpression vector using NdeI and XhoI restriction sites. Using the XhoI restriction site, the encoded protein does not contain a C-terminal His-tag.

### H-NS$_{Y61DM64D}$ expression vector
The vector pRD69 for expression of H-NS$_{Y61DM64D}$ was constructed by inserting a PCR fragment containing the *hns* gene mutated to encode aspartic acid instead of tyrosine/methionine at position 61 and 64 into pET3His using NdeI and XhoI restriction sites.

### Hha expression vector
The vector pRD38 for expression of N-terminally His tagged Hha was constructed by inserting a PCR fragment containing the *hha* gene into pET3His using XhoI and BamHI restriction sites.

### YdgT expression vector
The vector pRD39 for expression of N-terminally His tagged YdgT was constructed by inserting a PCR fragment containing the *ydgT* gene into pET3His using XhoI and BamHI restriction sites.

## Protein overproduction and purification

H-NS/H-NS$_{Y61DM64D}$/H-NS$_{E43A,E44A,S45A}$. BL21 (DE3) (RRID: WB-STRAIN: HT115(DE3)) $\Delta hns::kan$/frt pLysE (NT201, our lab) cells transformed with plasmids expressing H-NS/H-NS mutants were grown to an OD$_{600}$ of 0.4, and induced for 2 hr using IPTG (500 µM). For the H-N$_{E43A,E44A,S45A}$ protein, the cells were co-transformed with pRD252, coding for LacI to help suppress leaky expression of H-N$_{E43A,E44A,S45A}$. The cells were pelleted and lysed by sonication in 100 mM NH$_4$Cl, 20 mM Tris pH 7.2, 10% glycerol, 8 mM β-mercaptoethanol, 3 mM benzamidine). The soluble fraction was loaded onto a P11 column and eluted using a 100 mM-1 M NH$_4$Cl gradient, the protein eluted at 280 mM NH$_4$Cl. The peak fractions were dialysed to buffer B (identical to buffer A, but containing 130 mM NaCl instead of NH$_4$Cl) by overnight dialysis. The dialysate was loaded onto a heparin column (GE Healthcare) and eluted using a 130 mM-1 M NaCl gradient, the protein eluted at 350 mM NaCl. The pooled peak fractions were dialysed to buffer B and concentrated using a 1 ml Resource-Q column (GE Healthcare). The purity of the protein was verified on an SDS-PAGE gel. The protein concentration was determined using a Bicinchoninic Acid assay (Pierce BCA protein assay kit, Thermo Scientific).

## H-NS$_{1-58}$

BL21 (DE3) (RRID: WB-STRAIN: HT115(DE3)) $\Delta hns::kan$/frt pLysE (NT210, our lab) cells transformed with plasmids expressing H-NS/H-NS mutants were grown to an OD$_{600}$ of 0.4, and induced for 2 hr using IPTG (500 µM). The cells were pelleted and lysed by sonication in 100 mM NaCl, 20 mM Tris pH 7.2, 10% glycerol, 8 mM β-mercaptoethanol, 3 mM benzamidine). The soluble fraction was heated to 65°C for 10 min and then spun down at 10.000 RPM for 10 min. The supernatant was collected and a 1:1 ratio saturated ammonium sulfate (50 mM Tris pH 7.2, 4M ammonium sulfate) was gradually added to the cooled sample. The sample was spun down at 8.000 RPM for 15 min and a 1:1 ratio of 5 mM Tris pH 7.2, 15% glycerol was added to the supernatant. To remove further impurities, the sample was run through a 1 ml hydrophobic interaction column and 1 ml Blue-agarose column (the protein should not bind to either of these column) before finally binding the protein to 1 ml Resource-Q column (GE Healthcare). The protein was eluted with a 25 mM −1M gradient of NaCl, the protein eluted at roughly 380 mM NaCl. The purity of the protein was verified on an SDS-PAGE gel. The protein concentration was determined using a Bicinchoninic Acid assay (Pierce BCA protein assay kit, Thermo Scientific).

## Hha/YdgT

BL21 (DE3) (RRID: WB-STRAIN: HT115(DE3)) $\Delta hns$::frt, $hha::kan$, pLysE (our lab) cells transformed with plasmids pRD38/pRD39 expressing $hha/ydgT$ were grown at 37°C to an OD$_{600}$ of 0.4, and induced for two hours using IPTG (500 µM). The cells were pelleted and lysed in 20 mM HEPES pH 7.9, 1 M KCl, 10% glycerol, 8 mM β-mercaptoethanol. The soluble fraction was loaded onto a Ni-column. The column was first washed with buffer D (20 mM HEPES pH 7.9, 0.5 M KCl, 10% glycerol, 8 mM β-mercaptoethanol). The protein was then eluted using a 0 mM-0.5 M Imidazole gradient, the protein eluted at 300 mM Imidazole. The peak fractions were dialysed to buffer E (identical to buffer D, but containing 100 mM KCl) by overnight dialysis. The sample was then loaded onto pre-equilibrated SP Hi-Trap-column and Ni-column connected in series. After loading the samples on the column, the SP Hi-Trap -column was disconnected and the protein was eluted from the Ni-column using a 0–0.5 M imidazole gradient. The purity of the protein was verified on an SDS-PAGE gel. The protein concentration was determined using a Bicinchoninic Acid assay.

## Peptide production and purification

A truncated form of H-NS (H-NS$_{56-82}$) was synthesized by way of automated solid phase synthesis using standard protocols via Fmoc-strategy. Purification was performed by RP-HPLC with a Gemini 5µ C18 reversed phase column. Identity of the peptides was determined via MALDI-MS. The purity was determined by means of analytical RP-HPLC. The peptide was freeze dried and dissolved in 20 mM Tris pH 7.2, 300 mM KCl, 10% glycerol, 8 mM β-mercaptoethanol. The peptide concentration was determined using a Bicinchoninic Acid assay (Pierce BCA protein assay kit, Thermo Scientific).

## Size exclusion chromatography

Size exclusion chromatography was done using a Superose-12 column with a flow of 0.3 ml/min, pre-equilibrated with 10 mM Tris-HCl, 50 mM KCl, 5% glycerol containing or lacking 10 mM MgCl$_2$. The absorbance of the eluting fractions was measured at 215 nm. These experiments were performed in triplicate.

## DNA substrates

### DNA preparation

All experiments were performed using a random, AT-rich, 685 bp (32% GC) DNA substrate (*Laurens et al., 2012*). The DNA substrate was generated by PCR, and the PCR products were purified using a GenElute PCR Clean-up kit (Sigma-Aldrich). If required, DNA was $^{32}$P-labeled as described previously (*Wagner et al., 2011*).

## DNA-bridging assay

Streptavidin-coated paramagnetic Dynabeads M280 (Invitrogen) were washed once with 100 µL of 1xPBS and twice with Coupling Buffer (CB: 20 mM Tris-HCl pH 8.0, 2 mM EDTA, 2 M NaCl, 2 mg/mL BSA(ac), 0.04% Tween20) according to manufacturer instructions. After washing, the beads were

resuspended in 200 µL CB containing 100 nM biotinylated DNA. Next, the bead suspensions were incubated for 30 min on a rotary shaker (1000 rpm) at 25°C. After incubation, the beads were washed twice with Incubation buffer (IB: 10 mM Tris-HCl pH 8.0, 50 mM KCl, 10* mM MgCl$_2$, 5% v/v Glycerol, 1 mM DTT and 1 mM Spermidine) before resuspension in IB and addition of ±8000 cpm of radioactively labeled $^{32}$P 685 bp DNA. Radioactive DNA was supplemented with unlabeled 685 bp DNA to maintain a constant (20 nM) DNA concentration. The DNA-bridging protein H-NS (concentrations indicated in the text), and if applicable Hha or YdgT were added and the mixture was incubated for 30 min on a shaker (1000 rpm) at 25°C. To remove unbridged prey DNA, the beads were washed with IB, before resuspension in 12 µL stop buffer (10 mM Tris pH 8.0, 1 mM EDTA, 200 mM NaCl, 0.2% SDS). All samples were quantified through liquid scintillation counting over 10 min. All values recovered from the DNA-bridging assay were corrected for background signal (using a sample lacking H-NS), and normalized to a reference sample containing the amount of labeled $^{32}$P 685 bp DNA used in the assay. The samples were then run on a 5% 0.5x TBE gel to ensure DNA integrity. DNA bridging was calculated based on a reference sample containing 2 µL of prey DNA. All DNA-bridging experiments were performed in triplicate. Unless indicated otherwise all DNA-bridging experiments were performed in the presence of 10 mM of MgCl$_2$ and 3,3 µM of H-NS (10% more H-NS than is required for saturation - see *Figure 1A*). Each experiment contains 50 mM KCl, to which additional KCl or K-glutamate are added depending on the experimental condition tested.

## Tethered particle motion experiments

Tethered particle motion experiments were performed as reported previously (*Driessen et al., 2014*; *van der Valk et al., 2017*). Flow cells were prepared as described with minor modifications (*Driessen et al., 2014*; *van der Valk et al., 2017*). Here, before flowing in protein diluted in the experimental buffer (10 mM Tris-HCl pH 8.0, 50 mM KCl, 10 mM MgCl$_2$/EDTA, 5% v/v Glycerol, 1 mM DTT) the flow cell was washed using 4 flow cell volumes with the experimental buffer. Next, the flow cell was incubated for 10 min with protein solution before sealing the flow cell. The flow cell was maintained at a constant temperature of 25°C. Measurements were started 10 min after the introduction of protein solution. TPM experiments were done at least in duplicate. The data were analyzed as previously described (*Driessen et al., 2014*; *van der Valk et al., 2017*). The RMS was first converted to persistence length($L_p$) using *Equation 1* (*Göransson et al., 1990*):

$$RMS = 233 - \frac{156}{\left(1 + 0.08L_p\right)^{0.45}} \tag{1}$$

$L_p$ values were then used to calculate the fractional coverage (*Equation 2*) (*Göransson et al., 1990*):

$$n\vartheta = \frac{\sqrt{\frac{1}{L_{p,\,measured}}} - \sqrt{\frac{1}{L_{p,\,naked}}}}{\sqrt{\frac{1}{L_{p,\,saturated}}} - \sqrt{\frac{1}{L_{p,\,naked}}}} \tag{2}$$

The fractional coverage (*d*) was fit using the McGhee-von Hippel model for cooperative lattice binding (*Equation 3-5*) (*McGhee and von Hippel, 1974*):

$$\frac{\vartheta}{c} = K \cdot (1-d) \cdot \left(\frac{(2\omega+1)(1-d)+\vartheta-R}{(2\omega-1)(1-d)}\right)^{n-1} \cdot \left(\frac{1-(n+1)\vartheta+R}{2(1-d)}\right)^2 \tag{3}$$

where

$$R = \sqrt{(1-(n+1)\vartheta)^2 + 4\omega\vartheta(1-d)} \tag{4}$$

and

$$d = n\vartheta$$

Here, the association constant($K$) is described as a function of the protein concentration ($c$) and a cooperativity parameter ($\omega$).

To this end, weighted orthogonal distance regression (ODR) was performed to estimate the parameters of the nonlinear implicit equation describing the cooperative ligand binding. The binding site size (n) of H-NS was fixed to a value of 30 bp during regression, which corresponds to values determined previously (*Dame et al., 2006*; *Amit et al., 2003*). The association constant (K) and the cooperativity parameter (ω) were assumed to be positive real numbers. A custom fitting routine was implemented in Fortran and makes use of the ODRPACK library (*Boggs et al., 1992*).

## Molecular dynamics simulations

The starting conformation of the full-length H-NS dimer was constructed as described previously (*van der Valk et al., 2014*). The system was placed in a periodic dodecahedron box with a distance of at least 0.8 nm between the box edge and the most extended atom of the protein dimer, followed by the addition of water and ions. With this system we performed Molecular Dynamics (MD) simulations of full length H-NS at different concentrations of KCl, and $MgCl_2$ and with the addition of Hha, summing up to a total of six different systems. Hha was added by aligning the crystal structure containing the H-NS – Hha complex (PDB code 4ICG [*Ali et al., 2013*]) with the full-length structural model and copying the Hha molecules. System size ranged from 513457 atoms for the Hha systems to around 1,135,000 atoms for the other four systems.

Interactions between atoms were described by the AMBER99-SB-ILDN force field (*Lindorff-Larsen et al., 2010*), in combination with the TIP3P water model (*Jorgensen et al., 1983*). Long-range electrostatic interactions were treated via the Particle Mesh Ewald method (*Darden et al., 1993*; *Essmann et al., 1995*) with a short-range electrostatic cutoff distance at 1.1 nm. Van der Waals interactions were cut off at 1.1 nm. Preparation of the systems consisted of energy minimization equilibration of the solvent. Energy minimization was performed by the conjugate gradient method. After energy minimization, the positions of water molecules and ions were equilibrated by a 1 ns molecular dynamics run at a temperature of 298 K and a pressure of 1 bar in which the heavy atoms in the protein were position-restrained with a force constant in each direction of 1000 kJ/mol nm. After preparation, we performed 16 50 ns runs for each system, varying initial conditions by assigning new random starting velocities drawn from the Maxwell-Boltzmann distribution at 298 K. All simulations were performed with GROMACS v.4.6.3 (*Pronk et al., 2013*) at the Dutch National Supercomputer with the leap-frog integration scheme and a time step of 2 fs, using LINCS (*Hess et al., 1997*) to constrain all bonds. All simulations were performed in the isothermal-isobaric ensemble at a pressure of 1 bar, using the v-rescale thermostat (*Bussi et al., 2007*) and the Parrinello-Rahman barostat (*Parrinello and Rahman, 1981*).

Frames were stored every 10 ps. The first 10 ns of each simulation are excluded from analysis, unless stated otherwise. Analysis focused on determining contacts between domains, between H-NS and Hha, between H-NS and ions, and helical hydrogen bonds. Contact maps of interactions between residues in the H-NS dimer system were obtained by first calculating the minimum distance between each residue pair in the system. A residue pair is counted to be in contact if they are at a minimum distance of 0.6 nm or less. The probability of a contact is then calculated as the average over all 16 simulations (excluding the first 10 ns) and displayed as a contact probability matrix. We used a modified version of the g_mdmat tool in GROMACS (*Pronk et al., 2013*) in combination with Perl scripts to generate contact maps. To determine the location of ions with respect to the H-NS system, we calculated the minimum distance between each residue in the H-NS dimer and the ions and counted a contact if the distance between an H-NS residue and an ion is 0.6 nm or less. These contact probabilities ($P_{Mg2+}$, $P_{K+}$ and $P_{Cl-}$) are averaged over all 16 simulations and the two monomers. A similar procedure was performed to determine the probability of contacts between H-NS and Hha, $P_{Hha}$, and between H-NS domains, with $P_{1-40}$ indicating the probability of finding the DNA-binding domain (residues 96–137) close to the dimerization domain (residues 1–40). $P_{96-137}$ indicates the probability of finding the dimerization domain close to the DNA-binding domain.

To determine the location of the buckle in helix α3, we calculated the helical hydrogen bond distances $d_{O-N}$ for residues 22–67 in each monomer between the backbone carbonyl oxygen O of residue i and the backbone amide nitrogen N of residue i + 4. A hydrogen bond is counted to be in contact if they are at a minimum distance of 0.35 nm or less. These probabilities are averaged over all 16 simulations (excluding the first 10 ns) and the two monomers. Snapshots and movies were generated with PyMOL.

## Acknowledgements

We thank Bas de Mooij for his assistance in standardizing the bridging assay, Wim Jesse and Alexander Kros for synthesizing the H-NS$_{56-83}$ peptide and their assistance during its purification. We also thank Rosalie Driessen for her assistance with data analysis and all group members for valued discussions. JV acknowledges the use of the Dutch National Supercomputer Cartesius for the MD simulations.

## Additional information

### Funding

| Funder | Grant reference number | Author |
|---|---|---|
| NanonextNL of the Government of the Netherland and 130 partners | | Ramon A van der Valk<br>Geri F Moolenaar<br>Remus T Dame |
| Netherlands Organisation for Scientific Research | VIDI 864.08.001 | Ramon A van der Valk<br>Geri F Moolenaar<br>Nora Goosen<br>Remus T Dame |
| Netherlands Organisation for Scientific Research | Athena grant 700.58.802 | Jocelyne Vreede |
| Human Frontier Science Program | RGP0014/2014 | Andreas Hofmann<br>Remus T Dame |
| China Scholarship Council | No. 201506880001 | Liang Qin |
| Netherlands Organisation for Scientific Research | VICI 016.160.613 | Remus T Dame |

The funders had no role in study design, data collection and interpretation, or the decision to submit the work for publication.

### Author contributions

Ramon A van der Valk, Conceptualization, Formal analysis, Supervision, Validation, Investigation, Visualization, Methodology, Writing—original draft, Writing—review and editing; Jocelyne Vreede, Conceptualization, Resources, Data curation, Software, Formal analysis, Funding acquisition, Validation, Investigation, Visualization, Methodology, Writing—review and editing; Liang Qin, Formal analysis, Validation, Investigation, Visualization; Geri F Moolenaar, Nora Goosen, Conceptualization, Resources, Validation, Investigation, Methodology, Project administration; Andreas Hofmann, Software, Formal analysis, Investigation, Methodology; Remus T Dame, Conceptualization, Data curation, Formal analysis, Supervision, Funding acquisition, Visualization, Methodology, Writing—original draft, Project administration, Writing—review and editing

### Author ORCIDs

Remus T Dame iD http://orcid.org/0000-0001-9863-1692

### Decision letter and Author response

Decision letter https://doi.org/10.7554/eLife.27369.026
Author response https://doi.org/10.7554/eLife.27369.027

## Additional files

### Supplementary files

• Transparent reporting form
DOI: https://doi.org/10.7554/eLife.27369.025

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
