## [Decision Letter]

[Editors’ note: a previous version of this study was rejected after peer review, but the authors submitted for reconsideration. The first decision letter after peer review is shown below.]

Thank you for submitting your work entitled "Environment driven conformational changes modulate the function of H-NS" for consideration by *eLife*. Your article has been favorably evaluated by Kevin Struhl (Senior Editor) and three reviewers, one of whom is a member of our Board of Reviewing Editors. The reviewers have opted to remain anonymous.

Our decision has been reached after consultation between the reviewers. Based on these discussions and the individual reviews below, we regret to inform you that your work will not be considered for publication in *eLife*.

All three of the reviewers found the proposed model for the switch in H-NS binding modes to be interesting and important. However, they also thought that either a true functional (in vivo) experiment or a definitive test of the DNA-binding domain sequestration model would be required for publication in *eLife*. The reviewers hope their comments, particularly the very detailed and thoughtful comments of reviewer 3, will help you to improve the manuscript.

*Reviewer #1:*

The question of how the nucleoid protein H-NS is modulated by a range of different environmental signals has long been unanswered. In this study, van der Valk et al. developed and carried out novel methodologies to assay the impact of H-NS on DNA bridging and DNA stiffening at different concentrations of Mg^2+^ and K^+^ and in the presence of the H-NS co-regulators Hha and YdgT. The authors also examined the consequences of truncating H-NS and carried out molecular dynamic simulations of the effects of these conditions on the H-NS dimer. The results of these assays and modeling, led the authors to conclude that "H-NS switches between a "closed" and an "open", bridging-competent conformation driven by environmental cues and interaction partners. This model is extremely attractive, however in my opinion the authors have not tested their interesting ideas in sufficient depth. The study would be a more substantial contribution with additional experiments and discussion to address the following:

1) Many of the conclusions are based on molecular dynamic simulations. The authors should carry out independent experiments such as fluorescence quenching, to test the predictions of these simulations.

2) The authors examined the "modulation of DNA bridging by osmotic factors", but infer that the model is relevant to a range of environmental signals. This should be tested. As a minor point the authors should comment on how the concentrations of Mg^2+^ and K^+^ tested in vitro compare to the concentrations found to modulate H-NS activity in the cell.

3) The authors observe disparate effects of Hha and YdgT with respect to DNA stiffening. How do these differences relate to effects of Hha and YdgT in vivo? Can Hha really be a proxy for YdgT in the molecular simulations?

4) The authors make very general statements about the implications of their observations? Ideally, the contributions of filament formation versus bridging to the regulation of specific promoters should be examined.

*Reviewer #2:*

In this manuscript van der Valk and colleagues present evidence that certain ions and important regulatory factors from *E. coli* promote the ability of H-NS to bridge DNA molecules. The new in vitro DNA bridging assay they develop is elegant and is used to show that Mg and KCl above a certain concentration can promote DNA bridging by H-NS. The findings with Mg are consistent with the original findings from the Kenney and Yan labs. The authors go on to use the same in vitro assay to show that KCl and regulatory proteins that promote the activities of H-NS, including Hha and YdgT, also promote DNA bridging by H-NS. These findings are new. Finally, based on molecular dynamics simulations the authors suggest that Mg, K, and Hha influence bridging through a common mechanism. This proposed mechanism involves interacting directly with H-NS and blocking an otherwise inhibitory interaction between an oligomerization determinant and a DNA-binding determinant of the protein. The work is interesting but falls somewhat short. Additional biophysical experiments would be needed to strengthen the case that certain ions and proteins promote bridging by altering the distance between specific regions of H-NS. Furthermore, there are no in vivo or in vitro tests of the physiological relevance of Hha or YdgT's ability to promote bridging by H-NS.

1) For the H-NS (Y61D; M64D) double mutant that is predicted to be defective for dimer-dimer interactions, it would be important to show the EMSA analysis of its ability to bind DNA (currently mentioned as data not shown, subsection “The role of Mg^2+^ and H-NS multimerization in DNA bridging and DNA stiffening”, last paragraph). Ideally, the authors should also test whether this particular mutant is defective in vivo.

2) The modulation of DNA bridging and stiffening by truncated H-NS variants is interesting (subsection “Modulation of DNA bridging and DNA stiffening by truncated H-NS variants”). However, the authors should be careful about extrapolating the findings with these fragments of H-NS to other proteins such as H-NST. I think if the authors want to make strong claims about H-NST, then it would be better to directly test whether or not H-NST influences bridging (perhaps with the one from EPEC as it appears to have the most potent activity).

*Reviewer #3:*

Van der Valk et al. report new insights into the mechanism and regulation of DNA bridging by the bacterial transcriptional regulator H-NS. Specifically, the authors determined how magnesium, osmolarity, and Hha/YdgT modulate a switch between bridged H-NS filaments (H-NS filaments binding two distinct DNA segments) and stiffened H-NS filaments (N-HS binding to a single DNA segment). These two different modes of H-NS binding affect gene expression differently, and understanding both the physical basis for the switch and how it is regulated are crucial questions in the field of bacterial gene regulation. Van der Valk et al. propose and provide support for a novel hypothesis that the switch in binding modes involves sequestration or release of one of the two DNA-binding domains in an H-NS dimer by the N-terminal dimerization domain. Using a new, quantitative bridging assay, the authors report that magnesium and the H-NS modulators Hha/YdgT favor the bridging conformation whereas KCl and truncated H-NS derivatives that mimic other H-NS modulators decrease bridging at high concentrations. In contrast, stiffened filaments were only disrupted by interactions in the dimer-dimer interaction domain. Using molecular dynamics simulations to observe H-NS conformational changes with or without these factors, the authors found that a bulge in the central H-NS α-helix could facilitate interactions between the DNA-binding domain and the dimerization domain, resulting in a closed conformation that disfavors bridging. Factors that favor bridging appear to disfavor interactions between the DNA binding domain and dimerization domain, and results in a more open conformation. These results suggest a mechanism by which factors that influence H-NS filament conformations act by altering the sequestration of the DNA-binding domain.

The authors' findings significantly advance understanding of H-NS filament formation and its regulation, but their presentation in the submitted manuscript will be largely inaccessible to a general audience without revisions to the text and figures. Additionally, one relatively straightforward experiment would greatly increase the relevance of the findings to in vivo regulation. All these changes are achievable within a reasonable time and would make the manuscript suitable for high-profile publication.

1) Both the TPM and bridging assays need to be better explained in the text and figures. For starters, the ability of readers to understand each assay would be immensely aided by simply graphics that illustrate the assay and what is being measured. For the TPM assay, what is the physical difference in bead behavior between 0% and 100% coverage? This could be illustrated. For the bridging assay, what is the difference between no bridging and complete bridging? Is it just one H-NS dimer bridge between the two DNAs – or more than one bridge? This could be illustrated. For example, an earlier paper from this group provided an excellent illustration of the assay used (Figure 1 of Dame et al. 2006. Nature. 444:387). For both assays, what is the source of the DNA used (why was it chosen and where it is from)? Is the DNA from a gene naturally regulated by H-NS? These DNAs should be described, and perhaps shown as sequence in a supplemental figure, rather than referring readers to another paper.

The first paragraph of the subsection “The role of Mg^2+^ and H-NS multimerization in DNA bridging and DNA stiffening” should be rewritten to better explain the goals and set-up of the two assays, and the logic for why the experiments were carried out as described so that readers diverse backgrounds can follow the description.

2) One main conclusion is that Mg^2+^ ions interact with residues 22-35 of H-NS based on molecular dynamics. Do the authors believe this is principally an ionic interaction or some type of coordination of Mg^2+^ by these residues? Could this interaction be abolished by a mutation in H-NS? It would enhance the paper to discuss the properties of this interaction, but more experiments are not necessary. Is there precedence in the literature for Mg^2+^ interacting with proteins in a similar manner? If so, examples would be useful to cite and discuss.

a) The authors should also stipulate explicitly that the simulations were performed in the absence of DNA, and discuss whether the conclusions derived from simulations in the absence of DNA can be used to infer the behavior of H-NS in the presence of DNA. The current simulation provides evidence that the conformation of an H-NS dimer is dynamic in solution, but would this effect be different when H-NS is bound to DNA? A discussion of this question would make the results more complete.

b) Additionally, it is unclear why the authors' results in Figure 1 lead them to test the Mg^2+^ binding to H-NS. Did they favor this model because the high [Mg^2+^] does not inhibit DNA binding (subsection “The role of Mg^2+^ and H-NS multimerization in DNA bridging and DNA stiffening”, last paragraph)?

3) The authors propose that high K^+^ can disrupt the bridging of H-NS, which they relate to changes in the cell during osmotic stress. Although this observation is interesting, the authors should rule out the possibility that the high [Cl^-^] could destabilize H-NS DNA interaction rather than high [K^+^] destabilizing the bridging interaction, since it is known that Cl^-^ can destabilize protein-DNA interactions (Leirmo S, et al. 1987. Biochemistry 26:2095.). The authors should repeat the experiment in Figure 1 with potassium glutamate to test ionic conditions that actually correspond to the in vivo osmotic stress response. If K^+^ is responsible for the changes in bridging, then the potassium glutamate result will show a similar decrease in bridging as KCl.

4) The discussion on the Hha/YdgT effect on H-NS binding affinity is unclear because the experiment is not described well. The results are also confusing in that Hha and YdgT both increase bridging, but only Hha enhances binding affinity (by their assay). This result does not support the hypothesis put forth in the subsection “Modulation of H-NS by Hha and YdgT”, that both Hha and YdgT increase the binding affinity of H-NS. A clearer description of the results is needed to communicate the differences and similarities between the Hha and YdgT effects accurately. Additionally, a diagram illustrating how Hha and YdgT are proposed to affect filaments and a representation of this experiment also might improve the accessibility of the results. For Figure 3—figure supplement 1, including a graphical representation of coverage would immensely help in understanding the results.

a) It is surprising that Hha and H-NS together do not show cooperative DNA binding. The authors should discuss how these new results can be reconciled with previous publications showing that Hha (along with H-NS) is necessary for silencing of genes (e.g., Ali, S. S., et al. 2013. JBC. 288: 13356; Banos, R. C., et al. 2009. PLoS Genetics. 5:e1000513.) The results presented in those papers suggest that cooperative binding is necessary for forming a filament that can silence genes (a bridged filament), which would also suggest that filaments containing Hha and H-NS should cooperatively form a bridged filament. The result that Hha disrupts cooperative binding is inconsistent with that hypothesis, so the authors should discuss alternative mechanisms to explain how Hha and H-NS together silence genes.

[Editors’ note: what now follows is the decision letter after the authors submitted for further consideration.]

Thank you for resubmitting your work entitled "Environment driven conformational changes modulate H-NS DNA bridging activity" for further consideration at *eLife*. Your revised article has been evaluated by Kevin Struhl as the Senior Editor and Gisela Storz as the Reviewing Editor, and three reviewers.

There continues to be enthusiasm for your study, but after discussion, the reviewers feel the following additional revisions are critical for the study to live up to its potential as a significant advance.

1) Experimental tests of the predictions of the molecular dynamics simulations need to be carried out. This should involve mutating the residues (i.e. glutamates to serines) expected to interact with ions like Mg^2+^ and mutating residue R11 expected to destabilise the closed H-NS conformation. Assaying the effects of such mutations, using the experimental tools applied in Figure 1, should reveal differences in magnesium modulation of HNS binding as predicted by the simulations.

2) Downplay the hha/ydgT experiments, as they are not yet fully developed, and focus the paper on Mg^2+^ induced changes.

Each of the reviewers’ comments is also included below.

*Reviewer #1:*

In this revised manuscript van der Valk and colleagues present evidence that certain ions and important regulatory factors from *E. coli* promote the ability of H-NS to bridge DNA molecules. Key components of the manuscript include use of an elegant biochemical assay to demonstrate that Mg, as well as the regulatory factors Hha and YdgT promote DNA-bridging by H-NS. They also show that K inhibits DNA-bridging by H-NS, and that truncated derivatives of H-NS can modulate the ability of H-NS to bridge the DNA--findings that have implications for naturally occurring inhibitors of H-NS, such as H-NST. Based on molecular dynamics simulations the authors suggest that Mg and Hha promote bridging through a common mechanism. This proposed mechanism involves interacting directly with H-NS and blocking an otherwise inhibitory interaction between an oligomerization determinant and a DNA-binding determinant of the protein. In my earlier review I had indicated that additional biophysical experiments would be needed to strengthen the case that certain ions and proteins influence bridging by altering the distance between specific regions of H-NS. The authors didn't comment on this concern but indicate in their response to reviewer 1 that Alexa555 labelling of H-NS is problematic. Thus, a limitation of both the original and the current study is that there are no independent experimental tests of the predictions from the molecular dynamics simulations used to develop the highly interesting model the authors propose.

In my earlier review I had also indicated there are no in vivo or in vitro tests of the physiological relevance of Hha or YdgT's ability to promote bridging by H-NS. The authors didn't comment on my concern but indicate in response to reviewer 1 that they are conducting a separate single-molecule study to test the effects of different bridging efficiencies on RNA polymerase progression.

The authors adequately addressed my specific comments. In particular, they now provide in vitro analysis of the ability of HNS (Y61D;M64D) to bind DNA by EMSA. They also attempted to test the effects of H-NST (from EPEC) on bridging by H-NS in vitro, but ran into problems with H-NST solubility.

Additional independent experiments would strengthen the case that certain ions and proteins influence bridging by altering the distance between specific regions of H-NS.

*Reviewer #2:*

The paper by van der Valk, et al., systematically explores the effects of ions and accessory proteins on the multimerization and binding mode of H-NS on DNA. To do this they use a combination of a novel "bridging assay" (a pulldown of radioactive bait), molecular dynamics simulations, and TPM (tethered particle motion) assays.

Novel findings include that H-NS contains a "hinge" that allows it to go from an open to a closed conformation. In the closed conformation one of the DNA binding domains is proposed to be inaccessible because it interacts with the N-terminal "head" domain of the H-NS dimer. The primary determinants of whether the molecule is in an open or closed conformation are ions like K^+^ and Mg^2+^ as well as accessory molecules like Hha. This paper attempts to unify findings from many other previous publications into a single model.

I see the paper has been through at least one round of review already. I think it's in pretty good shape.

I'm not entirely convinced their models are correct regarding Hha vs. YdgT (see Figure 3). It makes little sense that by having *less* affinity for H-NS, that Hha in some way disrupts multimerization while YdgT (which has more affinity) promotes more multimerization, but of the closed H-NS conformer. They may be over-interpreting and making a model from limited data without thinking of alternatives. Furthermore, if Hha disrupts cooperativity, would not the HNS multimerization domain mutant (Y61DM64D) act the same as H-NS plus Hha? How does the mutant abolish bridging while Hha promotes it?

Another issue I have is that in their proposed "closed" conformation I see no way the central dimerization domain would be available for making an H-NS oligomer. The central dimerization domain would also be made inaccessible to binding a new dimer of H-NS. Perhaps they could explain it better?

I think the molecular dynamics simulations are interesting and they propose that certain residues in the protein interact with ions like Mg^2+^ to modulate the conformation of the molecule. It would be nice to see more simulations with "mutant" H-NS or even experimental data where the glutamates in question were changed to serine. I don't believe, however, those experiments are critical for publication.

*Reviewer #3:*

This is an interesting paper that adds important mechanistic detail to the known changes in H-NS:DNA complexes induced by a change in the availability of Mg^2+^, other divalent cations, and H-NS-like proteins.

Overall, this paper represents an important contribution to the field and I am supportive of publication. However, I think that the paper suffers in two ways and attempts could be made to resolve these issues.

First, the reliance on molecular dynamics is a potential weakness in my opinion. To me, this seems like an excellent way to generate a hypothesis that can be tested experimentally (see subsection “Mg^2+^ alters H-NS structure”) but this has not been done yet. Second, the paper attempts to deal with two issues; the role of cations and the role of interaction partners. As a result, I feel that the paper falls just short of "nailing" either story. In some ways, the Discussion reflects this and mostly ignores the interaction partners.

For me, it would be great to see a paper focused on cation interactions bolstered by a couple of simple experiments testing the predictions of the molecular dynamics. I appreciate that the paper has already been through one round of revision and so this idea may not be greeted enthusiastically by the authors or journal. With this in mind, it may be possible for the authors to rewrite the subsection “Modulation of DNA bridging and DNA stiffening by truncated H-NS variants” so that it is presented as a test of the molecular dynamics predictions. E.g. would some of these truncated H-NS proteins block some of the intramolecular interactions predicted by the dynamics?

---

## [Author Response]

[Editors’ note: the author responses to the first round of peer review follow.]

All three of the reviewers found the proposed model for the switch in H-NS binding modes to be interesting and important. However, they also thought that either a true functional (in vivo) experiment or a definitive test of the DNA-binding domain sequestration model would be required for publication in eLife. The reviewers hope their comments, particularly the very detailed and thoughtful comments of reviewer 3, will help you to improve the manuscript.Reviewer #1:The question of how the nucleoid protein H-NS is modulated by a range of different environmental signals has long been unanswered. In this study, van der Valk et al. developed and carried out novel methodologies to assay the impact of H-NS on DNA bridging and DNA stiffening at different concentrations of Mg^2+^ and K^+^ and in the presence of the H-NS co-regulators Hha and YdgT. The authors also examined the consequences of truncating H-NS and carried out molecular dynamic simulations of the effects of these conditions on the H-NS dimer. The results of these assays and modeling, led the authors to conclude that "H-NS switches between a "closed" and an "open", bridging-competent conformation driven by environmental cues and interaction partners. This model is extremely attractive, however in my opinion the authors have not tested their interesting ideas in sufficient depth. The study would be a more substantial contribution with additional experiments and discussion to address the following:1) Many of the conclusions are based on molecular dynamic simulations. The authors should carry out independent experiments such as fluorescence quenching, to test the predictions of these simulations.

Because of the delicate balance between the open and closed conformation of H-NS and the small size of the protein, it is not feasible to investigate this system using fluorescent tags, as these are far too invasive. Indeed, it has been reported earlier that fluorescently labeling H-NS with N-(1-pyrenyl)maleimide dramatically affects its functionality (Schroder et al., Biochem Biophys Res Comm 282, 2001). We have independently confirmed these findings and found that Alexa555 labeling of H-NS at C21 abrogates the DNA bridging capacity of H-NS, while it does not hamper DNA binding (confirmed by EMSA; see Author response image 1 – panel A) or DNA stiffening (confirmed by TPM; see Author response image 1 –panel B)).

**Author response image 1. respfig1:** DNA binding and activity of H-NS and H-NSC21Alexa555. (**A**) Electrophoretic mobility shift assay using a 32P labeled (AT-rich) curved DNA substrate as described in Dame et al., Bioch., 2001. (**B**) The root mean squared (RMS) displacement of a DNA tether bound by H-NS and H-NSC21Alexa555 as measured by TPM.

2) The authors examined the "modulation of DNA bridging by osmotic factors", but infer that the model is relevant to a range of environmental signals. This should be tested. As a minor point the authors should comment on how the concentrations of Mg^2+^ and K^+^ tested in vitro compare to the concentrations found to modulate H-NS activity in the cell.

Indeed we believe that the model is of generic value and could also explain the effect of other modulatory factors. It would be interesting to test additional factors, but we feel that this goes beyond the scope of the current studies, in which we aim to establish a framework using a series of well-defined and ‘simple’ modulatory factors. It is helpful to give the reader an impression of the concentrations of Mg^2+^ and K^+^ relevant to the cell. Whereas information on the effects of K^+^ in relation to the well-characterized osmoresponsive proU operon is known, little is currently known of the in vivoeffects of Mg^2+^ on H-NS functionality, beyond the observation that the Mg^2+^ stimulon (Minagawa et al., J. Bact, 2003) and some of the genes targeted by H-NS overlap. In order to describe the range of ion concentrations relevant in vivo, we have now included the following sentences in the Results section:

“The concentration range from 0 – 10 mM Mg^2+^ is considered to be physiologically relevant.”

And,

“This in vitroobservation mirrors the in vivoresponse of the ProU operon, at which KCl concentrations exceeding 100 mM are required to alleviate H-NS-mediated repression.”

3) The authors observe disparate effects of Hha and YdgT with respect to DNA stiffening. How do these differences relate to effects of Hha and YdgT in vivo? Can Hha really be a proxy for YdgT in the molecular simulations?

We agree that, given the high sequence identity and homology, the differences between Hha and YdgT in their ability to modulate H-NS function are striking. At this point it is not possible to correlate the distinct effects to in vivodata since data addressing the effects of Hha or YdgT alone on H-NS are unavailable. In the field, Hha and YdgT are considered interchangeable and their functions overlapping, but our data, for the first time, show that these small differences in sequence have substantial effects on how H-NS function is modulated. This observation may have important implications as it suggests that Hha- and YdgT-H-NS heteromers may have distinct target specificity.

4) The authors make very general statements about the implications of their observations? Ideally, the contributions of filament formation versus bridging to the regulation of specific promoters should be examined.

This issue has been addressed in the recent article by Kotlajich, M. V. et al., *eLife*, 2015. In their studies it was discovered that bridged complexes – and not filaments – reduce transcription by interfering with RNA polymerase progression. In our study we have shown that bridging efficiency – and not filament formation – is modulated by various regulatory factors. Thus our studies provide a rationale for the conditional impact of H-NS on transcription observed in Kotlajich et al. Currently, we are setting up single-molecule transcription experiments in which we can study the effect of different bridging efficiencies – obtained under different experimental conditions – RNA polymerase progression. The extent of these studies goes beyond the scope of the current work.

Reviewer #2:In this manuscript van der Valk and colleagues present evidence that certain ions and important regulatory factors from E. coli promote the ability of H-NS to bridge DNA molecules. The new in vitro DNA bridging assay they develop is elegant and is used to show that Mg and KCl above a certain concentration can promote DNA bridging by H-NS. The findings with Mg are consistent with the original findings from the Kenney and Yan labs. The authors go on to use the same in vitro assay to show that KCl and regulatory proteins that promote the activities of H-NS, including Hha and YdgT, also promote DNA bridging by H-NS. These findings are new. Finally, based on molecular dynamics simulations the authors suggest that Mg, K, and Hha influence bridging through a common mechanism. This proposed mechanism involves interacting directly with H-NS and blocking an otherwise inhibitory interaction between an oligomerization determinant and a DNA-binding determinant of the protein. The work is interesting but falls somewhat short. Additional biophysical experiments would be needed to strengthen the case that certain ions and proteins promote bridging by altering the distance between specific regions of H-NS. Furthermore, there are no in vivo or in vitro tests of the physiological relevance of Hha or YdgT's ability to promote bridging by H-NS.1) For the H-NS (Y61D; M64D) double mutant that is predicted to be defective for dimer-dimer interactions, it would be important to show the EMSA analysis of its ability to bind DNA (currently mentioned as data not shown, subsection “The role of Mg^2+^ and H-NS multimerization in DNA bridging and DNA stiffening”, last paragraph). Ideally, the authors should also test whether this particular mutant is defective in vivo.

We have included as supplementary information the results of an EMSA, which show that indeed the DNA binding capacity of the H-NS mutant is not severely affected (whereas binding is no longer cooperative). We have also attempted to introduce these two mutations in the endogenous *hns* gene with little success; during the λ red recombination procedure at the marker selection stage, the mutated gene repeatedly reverted to the wild type sequence. This suggests that the H-NS double mutant is not only defective in DNA compaction and organization, but perturbs cellular function in such a way that it severely decreases cell viability. This hypothesis is supported by random mutagenesis studies (Ueguchi, C. et al., J. Mol Biol, 1997), in which no mutations were found in the H-NS dimer-dimer interaction domain (a.a. residues 56-83), even though mutations were obtained throughout all other domains of H-NS.

2) The modulation of DNA bridging and stiffening by truncated H-NS variants is interesting (subsection “Modulation of DNA bridging and DNA stiffening by truncated H-NS variants”). However, the authors should be careful about extrapolating the findings with these fragments of H-NS to other proteins such as H-NST. I think if the authors want to make strong claims about H-NST, then it would be better to directly test whether or not H-NST influences bridging (perhaps with the one from EPEC as it appears to have the most potent activity).

We invested a large amount of time into cloning and purifying the natural truncated H-NS variant and H-NS modulator, H-NST (the EPEC variant), and tested this protein in our assays.

We initially focused on obtaining an untagged variant of this protein. As described earlier by Williamson and Free (MolMic, 2005), we were unable to obtain this protein in soluble form, despite our use of different types of mild detergents to enhance solubility. We therefore resorted to purifying and testing the soluble, tagged variant described by Williamson and Free, which in their hands somewhat reduced binding of H-NS to DNA in vitro. This tagged protein in our hands still suffers from solubility issues and were therefore only able to obtain it at low concentration.

Nevertheless, we tested the effect of adding this protein in the H-NS-DNA stiffening assay up to 8 µM tagged H-NST. In this range we observed no clear effect on DNA stiffness. Note that also with the H-NS_1-58_ peptide (which most resembles H-NST) the effect on DNA stiffness was mild, but significant in the range between 5-10 µM. Next, we investigated the effect of tagged H-NST on the H-NS-DNA bridging efficiency. In the bridging assay concentrations of up to 3 µM tagged H-NST can be achieved, while maintaining an identical buffer composition throughout the titration range. No significant reduction in bridging efficiency could be observed for tagged H-NST, whereas H-NS_1-58_ maximally perturbs H-NS mediated bridging at a concentration of 1 µM. The low perturbation efficiency of tagged H-NST in our assays mirrors the smaller effect of tagged H-NST compared to wtH-NST on H-NS mediated repression, observed in vivo by Williamson and Free.

The limited solubility of H-NST underscores the importance of using mimics with high solubility, such as the natural H-NS truncated derivatives used in our studies, to be able to investigate fundamental mechanistic aspects of H-NS activity crucial to its functional modulation.

We have softened the statement referred to by this reviewer by deleting the word ‘strongly’.

Reviewer #3:[…] The authors' findings significantly advance understanding of H-NS filament formation and its regulation, but their presentation in the submitted manuscript will be largely inaccessible to a general audience without revisions to the text and figures. Additionally, one relatively straightforward experiment would greatly increase the relevance of the findings to in vivo regulation. All these changes are achievable within a reasonable time and would make the manuscript suitable for high-profile publication.1) Both the TPM and bridging assays need to be better explained in the text and figures. For starters, the ability of readers to understand each assay would be immensely aided by simply graphics that illustrate the assay and what is being measured. For the TPM assay, what is the physical difference in bead behavior between 0% and 100% coverage? This could be illustrated. For the bridging assay, what is the difference between no bridging and complete bridging? Is it just one H-NS dimer bridge between the two DNAs – or more than one bridge? This could be illustrated. For example, an earlier paper from this group provided an excellent illustration of the assay used (Figure 1 of Dame et al. 2006. Nature. 444:387). For both assays, what is the source of the DNA used (why was it chosen and where it is from)? Is the DNA from a gene naturally regulated by H-NS? These DNAs should be described, and perhaps shown as sequence in a supplemental figure, rather than referring readers to another paper.The first paragraph of the subsection “The role of Mg^2+^ and H-NS multimerization in DNA bridging and DNA stiffening” should be rewritten to better explain the goals and set-up of the two assays, and the logic for why the experiments were carried out as described so that readers diverse backgrounds can follow the description.

We have added a figure to better illustrate how the assays used work (Figure 1—figure supplement 4). We have also modified the Materials and methods to include the following to further elucidate the minutia of the experiment: “and normalized to a reference sample containing the amount of labeled _32_P 685 bp DNA used in the assay.” The DNA substrate is now referred to as “a random, AT rich, 685 bp (32% GC) DNA substrate” in the subsection “DNA preparation”.

2) One main conclusion is that Mg^2+^ ions interact with residues 22-35 of H-NS based on molecular dynamics. Do the authors believe this is principally an ionic interaction or some type of coordination of Mg^2+^ by these residues? Could this interaction be abolished by a mutation in H-NS? It would enhance the paper to discuss the properties of this interaction, but more experiments are not necessary.

The simulations suggest that the interaction between Mg^2+^ and the glutamate residues in region 22-35 is mainly ionic, as the residence time of Mg^2+^ ions at those residues is relatively short, in the order of a few ns.

We altered the main text to include this information (lines 191 – 198):

“The likelihood of finding Mg^2+^ interacting with (i.e. being within 0.6 nm of) H-NS residues, indicated by PMg^2+^, revealed that Mg^2+^ has a preference for glutamate residues in region 22-35 (Figure 2), where the ions shield this region from interacting with the DNA binding domains. The magnesium ions transiently interact with the glutamate residues, with residence times in the order of a few ns (as illustrated in Figure 2—figure supplement 5).”

Is there precedence in the literature for Mg^2+^ interacting with proteins in a similar manner? If so, examples would be useful to cite and discuss.

We were unable to find any cases where Mg^2+^ interacts with proteins in a similar manner. However, we do suspect that similar interactions occur on other proteins, potentially modulating also their function.

a) The authors should also stipulate explicitly that the simulations were performed in the absence of DNA, and discuss whether the conclusions derived from simulations in the absence of DNA can be used to infer the behavior of H-NS in the presence of DNA. The current simulation provides evidence that the conformation of an H-NS dimer is dynamic in solution, but would this effect be different when H-NS is bound to DNA? A discussion of this question would make the results more complete.

This has now been addressed in the Results section where we have added:

“Even though the simulations were performed in absence of DNA, these observations indicate that DNA bridging is no longer possible in such a conformation, as the H-NS dimer can bind DNA only through its remaining/second DNA binding domain.”

b) Additionally, it is unclear why the authors' results in Figure 1 lead them to test the Mg^2+^ binding to H-NS. Did they favor this model because the high [Mg^2+^] does not inhibit DNA binding (subsection “The role of Mg^2+^ and H-NS multimerization in DNA bridging and DNA stiffening”, last paragraph)?

A motivation of testing the direct effect of Mg^2+^ on H-NS dimer structure has now been included:

“As Mg^2+^ does not affect the multimeric state of H-NS in solution (Figure 1—figure supplement 1), a model involving an effect on H-NS multimerization can be excluded. A structural effect of Mg^2+^ on individual units within H-NS filaments could explain the observed effects of Mg^2+^ on the bridging efficiency of H-NS.”

3) The authors propose that high K^+^ can disrupt the bridging of H-NS, which they relate to changes in the cell during osmotic stress. Although this observation is interesting, the authors should rule out the possibility that the high [Cl^-^] could destabilize H-NS DNA interaction rather than high [K^+^] destabilizing the bridging interaction, since it is known that Cl^-^ can destabilize protein-DNA interactions (Leirmo S, et al. 1987. Biochemistry 26:2095.). The authors should repeat the experiment in Figure 1 with potassium glutamate to test ionic conditions that actually correspond to the in vivo osmotic stress response. If K^+^ is responsible for the changes in bridging, then the potassium glutamate result will show a similar decrease in bridging as KCl.

This is an important point. We have carried out additional experiments to determine the effects of potassium-glutamate on DNA bridging and DNA stiffening by H-NS. The experiments reveal that potassium glutamate, similar to our findings with potassium chloride, reduces the DNA bridging efficiency, yet that higher concentrations of this salt are needed to achieve the same effects (see Figure 1—figure supplement 5). Throughout this concentration range DNA stiffening is only mildly affected (as seen with KCl), indicating that the intrinsic binding of H-NS to DNA is minimally affected by altered potassium glutamate concentrations. We have also investigated the effects of potassium-glutamate on the DNA binding affinity of H-NS. Here we found that, unlike our expectations based on literature (Leirmo et al., Biochemistry, 1987), potassium-glutamate reduces the DNA binding affinity of H-NS as seen in the figure below in panels C and D. The observation that despite a low DNA binding affinity DNA bridging is enhanced, supports our findings that the observed effects of ions on DNA bridging is direct, and mediated by a structural change in the protein. These findings further strengthen our conclusions that it is indeed DNA bridging and not DNA stiffening that is sensitive to environmental stimuli.

4) The discussion on the Hha/YdgT effect on H-NS binding affinity is unclear because the experiment is not described well. The results are also confusing in that Hha and YdgT both increase bridging, but only Hha enhances binding affinity (by their assay). This result does not support the hypothesis put forth in the subsection “Modulation of H-NS by Hha and YdgT”, that both Hha and YdgT increase the binding affinity of H-NS. A clearer description of the results is needed to communicate the differences and similarities between the Hha and YdgT effects accurately. Additionally, a diagram illustrating how Hha and YdgT are proposed to affect filaments and a representation of this experiment also might improve the accessibility of the results.

We have altered this part of the text to correct for this inadvertent omission. It now reads as follows:

“One possible explanation for the effects of Hha and YdgT, is that they effectively increase the DNA binding affinity of H-NS_47_. We find that the affinity of H-NS is significantly enhanced by Hha, but it is not significantly altered by YdgT (Figure 3—figure supplement 1).”

We have also added “Figure 3—figure supplement 2: Schematic depiction of the effects of Hha and YdgT.” To more clearly explain our findings with Hha and YdgT.

For Figure 3—figure supplement 1, including a graphical representation of coverage would immensely help in understanding the results.a) It is surprising that Hha and H-NS together do not show cooperative DNA binding. The authors should discuss how these new results can be reconciled with previous publications showing that Hha (along with H-NS) is necessary for silencing of genes (e.g., Ali, S. S., et al. 2013. JBC. 288: 13356; Banos, R. C., et al. 2009. PLoS Genetics. 5:e1000513.) The results presented in those papers suggest that cooperative binding is necessary for forming a filament that can silence genes (a bridged filament), which would also suggest that filaments containing Hha and H-NS should cooperatively form a bridged filament. The result that Hha disrupts cooperative binding is inconsistent with that hypothesis, so the authors should discuss alternative mechanisms to explain how Hha and H-NS together silence genes.

Indeed, earlier studies of H-NS and Hha clearly demonstrate functional interactions between Hha and H-NS required for gene silencing. However, these studies provide *direct* information only on the effect of Hha on the affinity of H-NS, and not on cooperativity of H-NS binding arising from the interaction with Hha. Altered cooperativity, i.e. an altered propensity to form protein-DNA filaments, is reported for the first time in our work as we are capable of accurately quantifying this aspect of H- NS binding. This was not possible with the methods employed by previous studies (such as those employed by Ali, S. S., et al. 2013. JBC. 288: 13356). We have updated the text and added an illustrative figure to explain how Hha lowers the cooperativity of H-NS DNA binding (Figure 3—figure supplement 2):

“The absence of cooperativity observed in the presence of Hha, indicates that Hha (unlike YdgT) interferes with filament formation. A higher effective affinity of H-NS- Hha to DNA compared to DNA bound H-NS-Hha may favor nucleation over filament formation (Figure 3—figure supplement 2).”

It is also important to note that there exists no discrepancy between our findings concerning the cooperativity of DNA binding and DNA bridging as these two occur at significantly different H-NS concentrations (150 nM H-NS for DNA binding, whereas bridging only begins at roughly 3 μM H-NS).

[Editors' note: the author responses to the re-review follow.]

There continues to be enthusiasm for your study, but after discussion, the reviewers feel the following additional revisions are critical for the study to live up to its potential as a significant advance.1) Experimental tests of the predictions of the molecular dynamics simulations need to be carried out. This should involve mutating the residues (i.e. glutamates to serines) expected to interact with ions like Mg^2+^ and mutating residue R11 expected to destabilise the closed H-NS conformation. Assaying the effects of such mutations, using the experimental tools applied in Figure 1, should reveal differences in magnesium modulation of HNS binding as predicted by the simulations.

In light of this suggestion we have cloned and purified several H-NS derivatives to test the occurrence of predicted structural changes in response to altered physico-chemical conditions:

a) H-NS_E42S, E43S, E44S_ (a mutant with α helix 3 predicted to be stable)

b) H-NS_E43A, E44A, S45A_ (another mutant with α helix 3 predicted to be stable)

c) H-NS_R11E_ (a mutation that should prevent the closed conformation by removing an important salt bridge)

We investigated these H-NS derivatives in both our DNA bridging and Tethered Particle Motion assays. These experiments yielded the following results:

a) H-NS_E42S, E43S, E44S_: No DNA binding is observed for this H-NS derivative, likely due to misfolding of the protein.

b) H-NS_E43A, E44A, S45A_: This H-NS derivative is capable of bridging DNA in the absence of Mg^2+^ (as seen in Figure 2—figure supplement 9), confirming both our MD simulations and our model

c) H-NS_R11E:_The intrinsic DNA binding affinity of this mutant is drastically reduced to such an extent that in the presence of Mg^2+^ no binding to DNA is observed (as seen in Author response image 2). This finding corroborates the findings of previous studies in vivo by Ueguchi et al.(Journal of Molecular Biology 1996) and in vitro by Bloch et al. (Nature Structural Biology 2003), in which it was shown that altering R11 perturbs H-NS function, likely by hampering its DNA binding activity.

**Author response image 2. respfig2:** DNA binding by the H-NS derivative H-NS_R11E_ measured by the Root Mean Square displacement of DNA.

2) Downplay the hha/ydgT experiments, as they are not yet fully developed, and focus the paper on Mg^2+^ induced changes.

We have modified the text accordingly.

Each of the reviewers’ comments is also included below.Reviewer #1:In this revised manuscript van der Valk and colleagues present evidence that certain ions and important regulatory factors from E. coli promote the ability of H-NS to bridge DNA molecules. Key components of the manuscript include use of an elegant biochemical assay to demonstrate that Mg, as well as the regulatory factors Hha and YdgT promote DNA-bridging by H-NS. They also show that K inhibits DNA-bridging by H-NS, and that truncated derivatives of H-NS can modulate the ability of H-NS to bridge the DNA--findings that have implications for naturally occurring inhibitors of H-NS, such as H-NST. Based on molecular dynamics simulations the authors suggest that Mg and Hha promote bridging through a common mechanism. This proposed mechanism involves interacting directly with H-NS and blocking an otherwise inhibitory interaction between an oligomerization determinant and a DNA-binding determinant of the protein. In my earlier review I had indicated that additional biophysical experiments would be needed to strengthen the case that certain ions and proteins influence bridging by altering the distance between specific regions of H-NS. The authors didn't comment on this concern but indicate in their response to reviewer 1 that Alexa555 labelling of H-NS is problematic. Thus, a limitation of both the original and the current study is that there are no independent experimental tests of the predictions from the molecular dynamics simulations used to develop the highly interesting model the authors propose.In my earlier review I had also indicated there are no in vivo or in vitro tests of the physiological relevance of Hha or YdgT's ability to promote bridging by H-NS. The authors didn't comment on my concern but indicate in response to reviewer 1 that they are conducting a separate single-molecule study to test the effects of different bridging efficiencies on RNA polymerase progression.The authors adequately addressed my specific comments. In particular, they now provide in vitro analysis of the ability of HNS (Y61D;M64D) to bind DNA by EMSA. They also attempted to test the effects of H-NST (from EPEC) on bridging by H-NS in vitro, but ran into problems with H-NST solubility.Additional independent experiments would strengthen the case that certain ions and proteins influence bridging by altering the distance between specific regions of H-NS.

In light of the reviewer’s suggestion we have designed an H-NS mutant that is predicted to directly affect the ability of H-NS to form the open and closed conformations.

In this mutant, we sought to abrogate the ability of H-NS to attain a closed conformation. In this light, we substituted the glutamic acids E43, E44, and serine S45 with Alanines to prevent the closed conformation. The DNA bridging assay (Figure 2—figure supplement 9) shows that this mutant indeed can bridge DNA in a Mg^2+^ independent manner. We note that there is a strong similarity in the Mg^2+^ dependence of the H-NS_E43A,E44A,S45A_ protein and wildtype H-NS in the presence of 4 µM YdgT. This similarity further supports our idea that YdgT and Hha promote the open conformation of H-NS to enhance its bridging efficiency.

Reviewer #2:[…] I'm not entirely convinced their models are correct regarding Hha vs. YdgT (see Figure 3). It makes little sense that by having less affinity for H-NS, that Hha in some way disrupts multimerization while YdgT (which has more affinity) promotes more multimerization, but of the closed H-NS conformer. They may be over-interpreting and making a model from limited data without thinking of alternatives. Furthermore, if Hha disrupts cooperativity, would not the HNS multimerization domain mutant (Y61DM64D) act the same as H-NS plus Hha? How does the mutant abolish bridging while Hha promotes it?

Indeed, as the reviewer summarizes, we find that Hha antagonizes the cooperativity inherent to H-NS DNA binding and multimerization. However, different from the results for the H-NS multimerization mutant (Y61DM64D), we find that H-NS + Hha is still capable of multimerization along the DNA (Figure 3). We agree with the referee that the disparate behaviour of the two related proteins, Hha and YdgT, is unexpected and hard to explain at a molecular mechanistic level based on the currently available information (unfortunately a H-NS-YdgT co-crystal structure is lacking). Therefore, we have significantly reduced the text in relation to our structural interpretation of these observations, and have removed the model in the accompanying figure.

Another issue I have is that in their proposed "closed" conformation I see no way the central dimerization domain would be available for making an H-NS oligomer. The central dimerization domain would also be made inaccessible to binding a new dimer of H-NS. Perhaps they could explain it better?

This is an astute question posed by the reviewer. We agree that the closed conformation in the shown simulation snapshots appears quite compact. It is important to realize that the structure is dynamic and that different physico-chemical conditions drive a bias within the population of proteins towards either open or closed conformation.

Interestingly, we found similar cooperativity in DNA binding by H-NS in the “open” conformation (H-NS_E43A,E44A,S45A_ as seen in Figure 3—figure supplement 1) and the “closed” conformation (wildtype H-NS in the absence of Mg^2+^ as seen in Figure 3—figure supplement 1). Only in the presence of Mg^2+^ does H-NS show a very high level of cooperative DNA binding (Figure 3—figure supplement 1). This indicates that cooperativity by H-NS is not promoted by either the “closed” or “open” conformations, but by the transition between the two states.

I think the molecular dynamics simulations are interesting and they propose that certain residues in the protein interact with ions like Mg^2+^ to modulate the conformation of the molecule. It would be nice to see more simulations with "mutant" H-NS or even experimental data where the glutamates in question were changed to serine. I don't believe, however, those experiments are critical for publication.

We agree with the reviewer that an experimental test of our structural model (and the specific residues of importance for the structural changes and Mg^2+^ sensing in the protein) significantly strengthens our observations. We have therefore designed and generated an H-NS derivative, in which the glutamic acids E43, E44, and S45 are substituted with Alanines. These mutations are predicted to prevent buckle formation and to result in a stable a-helix structure. This prevents the occurrence of the closed conformation, and is expected to result in a high efficiency of DNA bridging, which is not dependent on Mg^2+^. Our bridging assay (Figure 2—figure supplement 9) shows that this mutant indeed bridges DNA in a Mg^2+^ independent manner. See also our reply above.

Reviewer #3:This is an interesting paper that adds important mechanistic detail to the known changes in H-NS:DNA complexes induced by a change in the availability of Mg^2+^, other divalent cations, and H-NS-like proteins.Overall, this paper represents an important contribution to the field and I am supportive of publication. However, I think that the paper suffers in two ways and attempts could be made to resolve these issues.First, the reliance on molecular dynamics is a potential weakness in my opinion. To me, this seems like an excellent way to generate a hypothesis that can be tested experimentally (see subsection “Mg^2+^ alters H-NS structure”) but this has not been done yet. Second, the paper attempts to deal with two issues; the role of cations and the role of interaction partners. As a result, I feel that the paper falls just short of "nailing" either story. In some ways, the Discussion reflects this and mostly ignores the interaction partners.For me, it would be great to see a paper focused on cation interactions bolstered by a couple of simple experiments testing the predictions of the molecular dynamics. I appreciate that the paper has already been through one round of revision and so this idea may not be greeted enthusiastically by the authors or journal. With this in mind, it may be possible for the authors to rewrite the subsection “Modulation of DNA bridging and DNA stiffening by truncated H-NS variants” so that it is presented as a test of the molecular dynamics predictions. E.g. would some of these truncated H-NS proteins block some of the intramolecular interactions predicted by the dynamics?

We have taken the reviewers comment to heart, but we fear that this is a misrepresentation of our experiments with truncated H-NS proteins. Instead we chose to directly test the model described by the MD simulations by generating an H-NS mutant predicted to persist in the “open” conformation. In this light, we substituted the glutamic acids E43, E44, and S45 with Alanines to prevent the closed conformation. Our bridging assay (Figure 2—figure supplement 9) shows that this mutant indeed bridges DNA in a Mg^2+^ independent manner.